# Sources of gene expression variation in a globally diverse human cohort

Dylan J. Taylor[1], Surya B. Chhetri[2,3], Michael G. Tassia[1], Arjun Biddanda[1], Stephanie M. Yan[1], Genevieve L. Wojcik[4], Alexis Battle[2,5,6,7] & Rajiv C. McCoy[1]✉

Genetic variation that influences gene expression and splicing is a key source of phenotypic diversity[1–5]. Although invaluable, studies investigating these links in humans have been strongly biased towards participants of European ancestries, which constrains generalizability and hinders evolutionary research. Here to address these limitations, we developed MAGE, an open-access RNA sequencing dataset of lymphoblastoid cell lines from 731 individuals from the 1000 Genomes Project[6], spread across 5 continental groups and 26 populations. Most variation in gene expression (92%) and splicing (95%) was distributed within versus between populations, which mirrored the variation in DNA sequence. We mapped associations between genetic variants and expression and splicing of nearby genes (cis-expression quantitative trait loci (eQTLs) and cis-splicing QTLs (sQTLs), respectively). We identified more than 15,000 putatively causal eQTLs and more than 16,000 putatively causal sQTLs that are enriched for relevant epigenomic signatures. These include 1,310 eQTLs and 1,657 sQTLs that are largely private to underrepresented populations. Our data further indicate that the magnitude and direction of causal eQTL effects are highly consistent across populations. Moreover, the apparent 'population-specific' effects observed in previous studies were largely driven by low resolution or additional independent eQTLs of the same genes that were not detected. Together, our study expands our understanding of human gene expression diversity and provides an inclusive resource for studying the evolution and function of human genomes.

Genetic variation that affects gene expression and splicing accounts for a large proportion of phenotypic differences within and between species[1]. By correlating patterns of expression and splicing with variation at the level of DNA, past research has helped reveal the genetic basis of these molecular traits and their relationships with higher-order phenotypes[2–5]. Previous molecular association studies in humans have been strongly biased towards individuals of European ancestries, which potentially constrains generalizability and hinders our understanding of human gene expression diversity and evolution[7–9]. Research has also demonstrated that the inclusion of diverse samples breaks up linkage disequilibrium (LD), which improves resolution for identifying causal variants[10].

Motivated by these points, several studies have profiled gene expression in geographically diverse samples[11–13]. These studies have generally observed that gene expression and splicing differences between populations are rare and that divergence in these molecular phenotypes does not clearly reflect patterns of population divergence. Studies have also revealed an abundance of genetic variants associated with levels of gene expression (termed eQTLs). Promoter proximal eQTLs possessed larger effects, on average, and tended to be shared across populations[12]. Although foundational, these studies were generally characterized by small sample sizes and/or assayed gene expression using microarrays. This limits statistical power and resolution for molecular QTL mapping and hinders integration and comparison with modern sequencing-based datasets. Meanwhile, recent work by consortia such as MESA, GALA II and SAGE have generated RNA sequencing (RNA-seq) data from thousands of samples and include representation from African American and Latin American populations[14,15], but their controlled access poses barriers to re-use, and in some cases are restricted to disease-related research.

To address this gap, here we developed MAGE, an open resource for multi-ancestry analysis of gene expression. MAGE comprises RNA-seq data from a large sample of lymphoblastoid cell lines (LCLs) derived from individuals across geographically diverse human populations. Using these data, we performed the following analyses: (1) quantified the distribution of gene expression and splicing diversity; (2) mapped genetic variation that influences gene expression and splicing at high resolution; and (3) examined the evolutionary forces that shape such variation and the causes of apparent heterogeneity in its effects across populations. Together, our work offers a more complete view of the magnitude, distribution and genetic sources of human gene expression and splicing diversity.

[1]Department of Biology, Johns Hopkins University, Baltimore, MD, USA. [2]Department of Biomedical Engineering, Johns Hopkins University, Baltimore, MD, USA. [3]Center for Computational Biology, Johns Hopkins University, Baltimore, MD, USA. [4]Department of Epidemiology, Johns Hopkins University, Baltimore, MD, USA. [5]Department of Computer Science, Johns Hopkins University, Baltimore, MD, USA. [6]Department of Genetic Medicine, Johns Hopkins University, Baltimore, MD, USA. [7]Malone Center for Engineering in Healthcare, Johns Hopkins University, Baltimore, MD, USA. ✉e-mail: rajiv.mccoy@jhu.edu

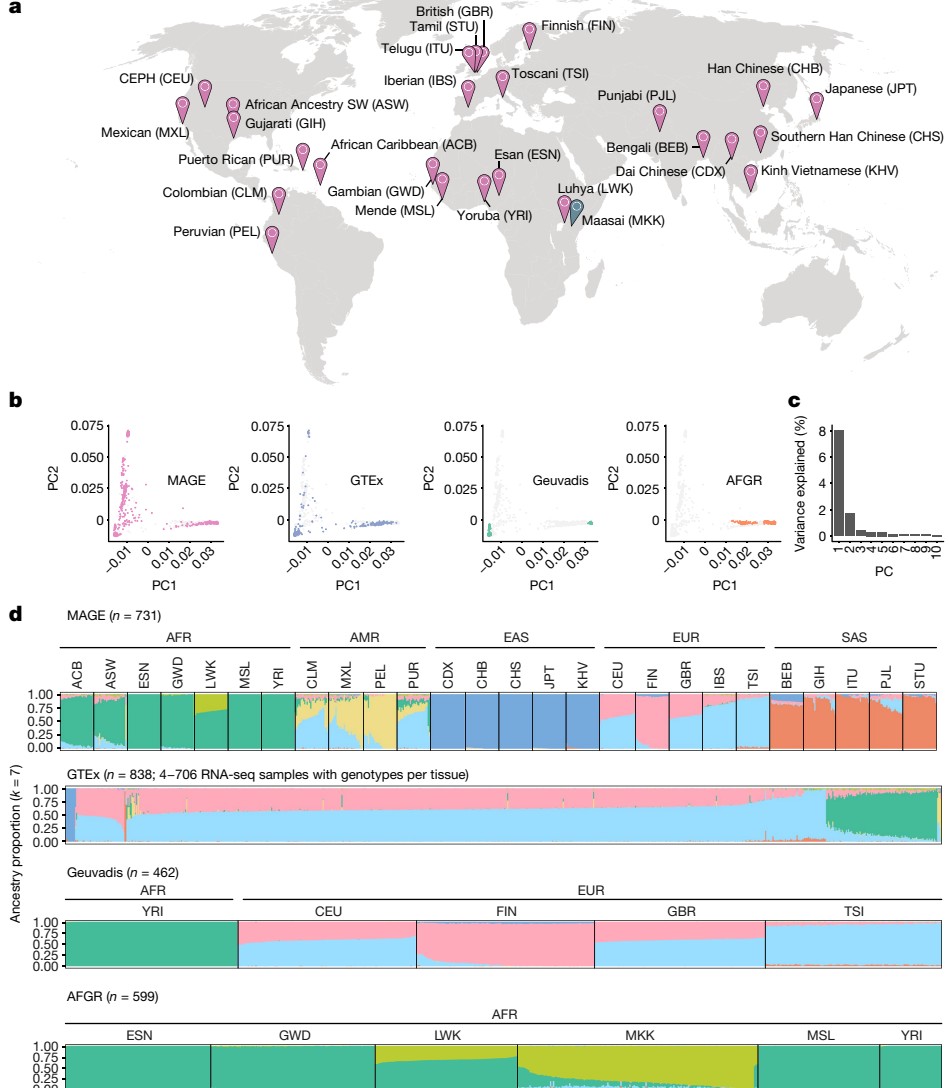

**Fig. 1 | A globally diverse transcriptomics dataset. a**, RNA-seq data were generated from LCLs from 731 individuals from the 1KGP[6], roughly evenly distributed across 26 populations and 5 continental groups. Populations included in MAGE are indicated in pink, whereas the Maasai population is in blue as it is present in the AFGR[17] dataset (based on sequencing of HapMap[57] cell lines) but not in the 1KGP or MAGE. Full population descriptors can be found at https://catalog.coriell.org/1/NHGRI/About/Guidelines-for-Referring-to-Populations. **b**, Genotype principal component 1 (PC1) and PC2 comparing MAGE to other large studies with paired RNA and whole-genome sequencing data. Samples from the specified study (that is, MAGE, Geuvadis, GTEx and AFGR) are depicted with coloured points, whereas samples from other studies are depicted with grey points in each respective panel. **c**, Proportion of variance explained by the first ten PCs. **d**, ADMIXTURE[58] results displaying proportions of individual genomes (columns) attributed to inferred ancestry components. For MAGE, Geuvadis[4] and AFGR, samples are stratified according to population and continental group labels from the respective source projects, whereas GTEx[26] does not include population labels. A subset of 1KGP samples are present across multiple RNA-seq studies and therefore appear in multiple panels, but these samples were not duplicated within the input to ADMIXTURE. Ancestry components are modelling constructs that do not directly correspond to true ancestral populations, and the results of ADMIXTURE analysis strongly depend on sampling characteristics of the input data. Although $k = 7$ minimizes the cross-validation error within this combined dataset (Supplementary Fig. 1), alternative choices of $k$ reflect structure at different scales (Supplementary Fig. 2). Map in **a** adapted from the US CIA World Factbook, 2005.

## A multi-ancestry RNA-seq resource

We performed RNA-seq of LCLs from 731 individuals from the 1000 Genomes Project[6] (1KGP), which represented 26 globally distributed populations (27–30 individuals per population) across 5 continental groups (Fig. 1a). Although we emphasize the greater genetic diversity within African populations—a point obscured by ADMIXTURE analysis and principal component analysis (PCA)[16]—these visualizations demonstrate that our study includes data from several non-African ancestry groups that were poorly represented in previous studies (Fig. 1b–d and Supplementary Figs. 2 and 3; also see the African Functional Genomics Resource (AFGR)[17]). All 731 samples were sequenced in a single laboratory across 17 batches, and sample populations were stratified across batches to avoid confounding between population and batch (Supplementary Fig. 5). We quantified gene expression levels using gene annotations from GENCODE (v.38) and used an annotation agnostic approach implemented by LeafCutter[18] to quantify alternative splicing patterns (Supplementary Methods and Supplementary Fig. 4).

## Gene expression and splicing diversity

The majority of variation in DNA sequence is distributed within as opposed to between human populations[19,20]. Previous studies have explored the extent to which this pattern holds for gene expression

diversity, finding that population labels explain 3–25% of the total variation in gene expression[4,13]. However, these studies were limited by either sample size or diversity, motivating our analysis within MAGE.

To this end, we fit a linear model relating the expression level of each gene with continental group and population labels from the 1KGP. After regressing out sequencing batch and sex effects, continental group explained an average of 2.92% of variance in gene expression level across tested genes (s.d. = 3.18%), whereas population label explained an average of 8.40% of variance (s.d. = 4.43%; Fig. 2a). Although small, these proportions exceed null expectations assuming no population structure (one-tailed permutation test: $P_{\text{continental group}}$, $P_{\text{population}} < 1 \times 10^{-3}$). Notably, the proportion of variance explained was smaller, on average, than reported in a previous study that included samples from the San population, whose ancestors diverged (with subsequent gene flow) from other populations in the dataset >100,000 years ago[13,21].

We observed similar patterns for alternative splicing, whereby—after regressing out technical variation—continental group and population explained an average of only 1.23% (s.d. = 1.93%) and 4.58% (s.d. = 2.24%) of variance, respectively (Fig. 2c; one-tailed permutation test: $P_{\text{continental group}}$, $P_{\text{population}} < 1 \times 10^{-3}$). The proportions of variance in gene expression and splicing explained by population label are not directly comparable because of differences in their units of measurement. However, our observations are qualitatively consistent with previous reports that expression level varies more between populations than splicing[13].

Notably, we also observed that within-population variance in expression (one-tailed analysis of deviance: $\chi^2$ (4, $N = 100,890$) = 17,623, $P < 1 \times 10^{-10}$) and splicing (one-tailed analysis of deviance: $\chi^2$ (4, $N = 164,335$) = 1550.6, $P < 1 \times 10^{-10}$) differed among continental groups. That is, there were higher average variances (across all tested genes) observed within the African continental group compared with the Admixed American continental group (Fig. 2b,d and Supplementary Fig. 7). These results are consistent with the demonstrated decline in genetic diversity resulting from serial founder effects during human global migrations[22,23]. Although significant, the magnitudes of these differences in variances were smaller than the magnitude of the decline in genetic diversity, which probably reflect the non-genetic environmental and stochastic contributions to gene expression and splicing variance that similarly affect all samples.

## Genetic effects on gene expression

### Mapping eQTLs and sQTLs at high resolution

MAGE offers a valuable resource for uncovering the genetic factors that drive variation in gene expression and splicing, including genetic variation that is largely private to historically underrepresented populations. By intersecting published genotype data from the same set of samples[24], we mapped cis-eQTLs and cis-sQTLs within 1 Mb of the transcription start site (TSS) of each gene. We define eGenes and sGenes as genes with an eQTL or sQTL, respectively, and eVariants and sVariants as the individual genetic variants defining an eQTL or sQTL signal, respectively. We note that although we performed QTL mapping for genes on the autosomes and the X chromosome, we focus on results from the autosomes here owing to several methodological details that are specific to the X chromosome (Supplementary Methods). Across 19,539 autosomal genes that passed expression-level filtering thresholds (Supplementary Methods), we discovered 15,022 eGenes and 1,968,788 unique eVariants (3,538,147 significant eVariant–eGene pairs; 5% false discovery rate (FDR)). Additionally, across 11,912 autosomal genes that passed splicing-filtering thresholds, we discovered 7,727 sGenes and 1,383,540 unique sVariants (2,416,177 significant sVariant–sGene pairs; 5% FDR).

The inclusion of genetically diverse samples in association studies reduces the extent of LD and improves mapping resolution[8,10] (Supplementary Fig. 11). With this advantage in mind, we used SuSiE[25] to

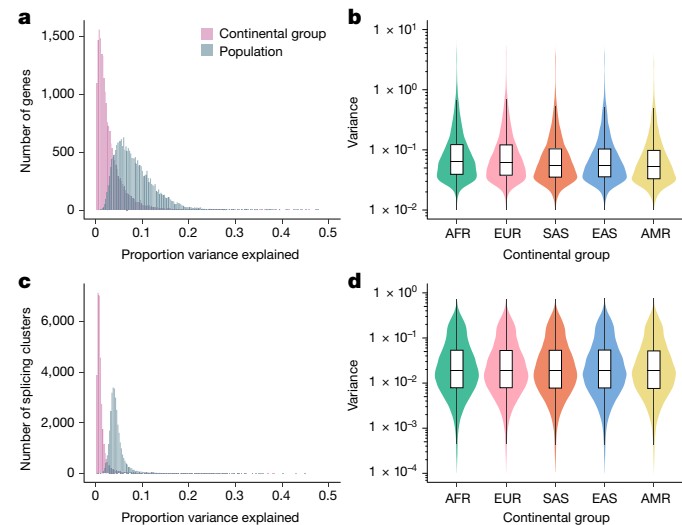

**Fig. 2 | Patterns of transcriptomic diversity within and between populations. a**, Per gene estimates of the proportion of variance in gene expression level that is partitioned between continental groups and populations, as opposed to within continental groups or populations. **b**, Variance in expression level per gene (*n* = 20,154 genes) differs across continental groups, consistent with underlying differences in levels of genetic variation. Variance in expression level differs between continental groups ($P < 1 \times 10^{-10}$, one-tailed analysis of deviance). **c**, Per splicing cluster estimates of the proportion of variance in alternative splicing (intron excision ratios) that is partitioned between continental groups and populations, as opposed to within continental groups or populations. **d**, Variance in alternative splicing (intron excision ratios) per splicing cluster (*n* = 32,867 splicing clusters) differs across continental groups, consistent with underlying differences in levels of genetic variation. Variance in splicing differs between continental groups ($P < 1 \times 10^{-10}$, one-tailed analysis of deviance). In **b** and **d**, bars represent the first, second (median) and third quartiles of the data and whiskers are bound to 1.5× the interquartile range.

perform fine mapping for all eGenes and the introns of all sGenes to identify causal variants that drive each QTL signal. For each gene and intron, SuSiE identifies one or more credible sets, representing independent causal eQTL and sQTL signals and whereby each credible set contains as few variants as possible while maintaining a high probability of containing the causal variant. To obtain a gene-level summary of the sQTL fine-mapping results, we collapsed intron-level credible sets into gene-level credible sets by iteratively merging intron-level credible sets for each sGene (Supplementary Methods). We identified at least one credible set for 9,807 (65%) eGenes and 6,604 (85%) sGenes, which we define as fine-mapped eGenes and sGenes, respectively. Consistent with previous results[4,26,27], we observed widespread allelic heterogeneity across fine-mapped genes, with 3,951 (40%) of fine-mapped eGenes and 3,490 (53%) of fine-mapped sGenes exhibiting more than one distinct credible set (Fig. 3a and Extended Data Fig. 2c). We also achieved high resolution in identifying putative causal variants that drive expression changes. That is, of 15,664 eQTL credible sets, 3,992 (25%) contained a single variant (median 5 variants per credible set; mean = 15.8, s.d. = 65.7; Fig. 3b). Similarly, for sQTLs, 3,569 out of 16,451 (22%) credible sets contained a single variant (median 7 variants per credible set; mean = 23.6, s.d. = 99.1; Extended Data Fig. 2d). For downstream analyses, we selected a single representative 'lead QTL' from each eGene and sGene gene-level credible set.

For each lead eQTL, we calculated its effect size using an implementation of the allelic fold change (aFC)[28] statistic that quantifies eQTL effect sizes conditional on all other lead eQTLs for that gene (Supplementary Methods). We observed that 2,031 (13%) lead eQTLs had a greater than twofold effect on gene expression (median $|\log_2(\text{aFC})| = 0.30$; mean = 0.51, s.d. = 0.64; Extended Data Fig. 1). This was a slightly smaller

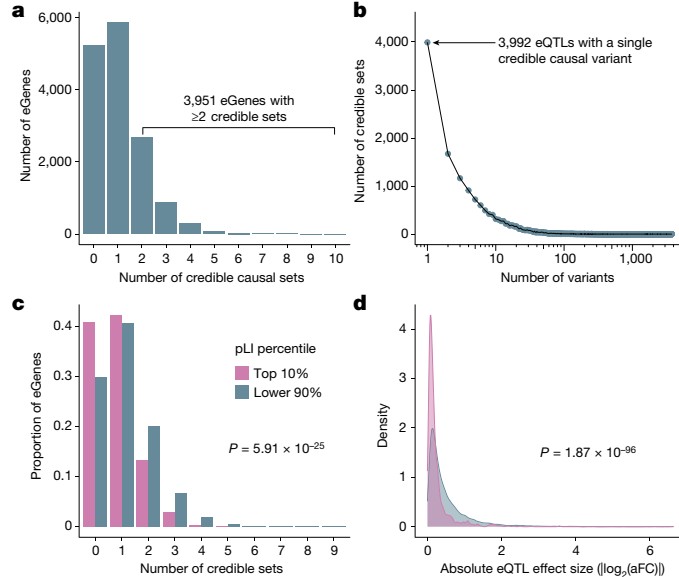

**Fig. 3 | Mapping high-resolution eQTLs. a**, Number of credible sets per eGene, demonstrating evidence of widespread allelic heterogeneity, whereby multiple causal variants independently modulate expression of the same genes. **b**, Fine-mapping resolution, defined as the number of variants per credible set. **c**, A signature of negative selection against expression-altering variation, whereby eGenes under strong evolutionary constraint (defined as the top pLI decile reflecting intolerance to loss-of-function mutations; pink) possess fewer credible sets, on average, than other genes (blue). **d**, A signature of negative selection against expression-altering variation, whereby eQTLs of genes under strong evolutionary constraint (top pLI decile; pink) have smaller average effect sizes (aFC) than other genes (blue).

proportion than previously reported by GTEx[26], but we propose that this is partially explained by the small sample sizes in some GTEx tissues, which drives a stronger 'winner's curse', whereby effects are systematically overestimated[29].

## Evidence of selective constraint

Previous studies of large population cohorts have identified sets of genes under strong mutational constraint, whereby negative selection has depleted loss-of-function point mutations and copy number variation[30]. One metric for quantifying mutational constraint on genes is the probability of intolerance to loss-of-function mutations (pLI)[30]. In our data, we observed that eGenes possessed significantly lower mean pLI scores (mean = 0.261, s.d. = 0.395) than non-eGenes (mean = 0.304, s.d. = 0.409; two-tailed Wilcoxon rank-sum test: $W = 11,596,590$, $P = 3.89 \times 10^{-7}$). Additionally, highly constrained eGenes (top 10% of pLI) tended to possess fewer credible sets (mean = 0.80, s.d. = 0.82) than other eGenes (mean = 1.12, s.d. = 1.04; two-tailed quasi-Poisson generalized linear model: $\hat{\beta} = -0.354$, $P = 5.91 \times 10^{-25}$; Fig. 3c). Moreover, the average effect size of lead eQTLs within highly constrained genes (mean $|\log_2(\text{aFC})| = 0.25$; s.d. = 0.36) was smaller than that of other genes (mean $|\log_2(\text{aFC})| = 0.53$; s.d. = 0.65; two-tailed Wilcoxon rank sum test: $W = 3,789,053$, $P = 1.87 \times 10^{-96}$; Fig. 3d). This difference was apparent regardless of whether the minor allele is associated with higher ($\Delta$mean $|\log_2(\text{aFC})| = -0.277$; two-tailed Wilcoxon rank-sum test: $W = 928,592$, $P = 1.39 \times 10^{-50}$) or lower expression ($\Delta$mean $|\log_2(\text{aFC})| = -0.268$; two-tailed Wilcoxon rank sum test: $W = 967,228$, $P = 2.97 \times 10^{-47}$), consistent with a model of stabilizing selection whereby gene expression is maintained within an optimal range. These results indicate an association between constraint against loss-of-function protein-coding sequence variation (that is, pLI) and constraint against expression-altering variation (that is, number of credible sets and eQTL effect sizes). This association held for several other metrics of

mutational constraint that include intolerance to copy number variation (that is, pHaplo and pTriplo) as well as divergence-based estimates of sequence conservation in putative promoter elements (Extended Data Fig. 3). Together, our results are consistent with previous analyses demonstrating weak, but measurable, selection against expression-altering variation[31].

## Functional enrichment of QTLs

Taking advantage of the high resolution of putative causal signals, we quantified the enrichment of fine-mapped lead eQTLs in 15 predicted chromatin-state annotations across 127 reference epigenomes from the Roadmap Epigenomics chromHMM model[32]. Enrichment was most pronounced within promoter regions, specifically at active TSSs (TssA) and flanking regions (TssAFlnk), but modest enrichments were also apparent within enhancer regions (Enh and EnhG), especially for blood cell types (Fig. 4a and Supplementary Fig. 12B). Conversely, quiescent, repressive and heterochromatic regions were depleted of eQTLs. We further extended our analysis to primary DNase hypersensitivity site (DHS) annotations, and we observed a strong enrichment of lead eQTLs in DHSs of blood and T cell samples (Supplementary Fig. 12C).

Focusing on data from LCLs, we next explored the relationship between epigenomic enrichments and eQTL effect sizes ($|\log_2(\text{aFC})|$). Promoter-associated enrichment was consistent across eQTL effect size deciles, and enrichment within poised regulatory regions such as bivalent TSS (TSSBiv) and bivalent enhancers (EnhBiv) was most apparent for eQTLs of large effect sizes (Supplementary Fig. 13A,B). By contrast, eQTLs located within chromatin states associated with transcribed regions (Tx, TxWk and TxFlnk) predominantly exhibited lower effect sizes (Supplementary Fig. 13C). These qualitative trends were replicated in other primary blood cell types (Supplementary Figs. 14–17). Using additional DHS-based annotations from Roadmap Epigenomics[32], we observed larger median eQTL effect sizes in promoter regions relative to enhancers and dyadic (that is, acting as both promoter and enhancer) regions (Fig. 4b). This pattern was similarly replicated across other primary blood-related cell types (Supplementary Figs. 14–17). Using chromatin immunoprecipitation followed by sequencing data from ENCODE[33], we also observed that lead eQTLs were significantly enriched within 312 (92.30%; Bonferroni-adjusted $P < 0.05$) transcription factor (TF) binding sites, including canonical promoter-associated TFs such as POLR2A, TAF1, JUND, ATF2 and KLF5, as well as TFs such as HDACs, EP300 and YY1, which are typically associated with enhancers (Supplementary Fig. 12A).

We also investigated the genomic context of our fine-mapped *cis*-sQTLs. We observed strong enrichment of lead sQTLs in several key splicing-relevant annotations, including splice donor sites ($\log_2(\text{fold enrichment}) = 6.07$, 95% confidence interval (CI) = 4.09–8.04) splice acceptor sites ($\log_2(\text{fold enrichment}) = 5.52$, 95% CI = 3.54–7.50) and nearby regions ($\log_2(\text{fold enrichment}) = 4.15$, 95% CI = 3.70–4.62) at intron–exon boundaries (Fig. 4c). Despite their magnitude of enrichment, variants in canonical splice sites and splice regions represented a minority of lead sQTLs, with a greater abundance of sQTLs falling within 5′ and 3′ untranslated regions (UTRs), as well as exons of both coding and noncoding genes. Although exhibiting weaker enrichments, these annotation categories together covered a much larger mutational target size and may encompass splicing enhancers and cryptic splice sites. By contrast, intergenic regions were strongly depleted of lead sQTLs ($\log_2(\text{fold enrichment}) = -2.51$, 95% CI = −2.58 to −2.43). Together, these findings provide support for the biological validity of the fine-mapped *cis*-QTLs and insight into the mechanisms by which these variants affect gene expression and splicing.

## Colocalization of eQTLs and sQTLs and GWAS hits

To explore the role of expression-associated genetic variation in human complex traits, we next sought to discover shared signals between fine-mapped MAGE *cis*-eQTLs and *cis*-sQTLs and results from

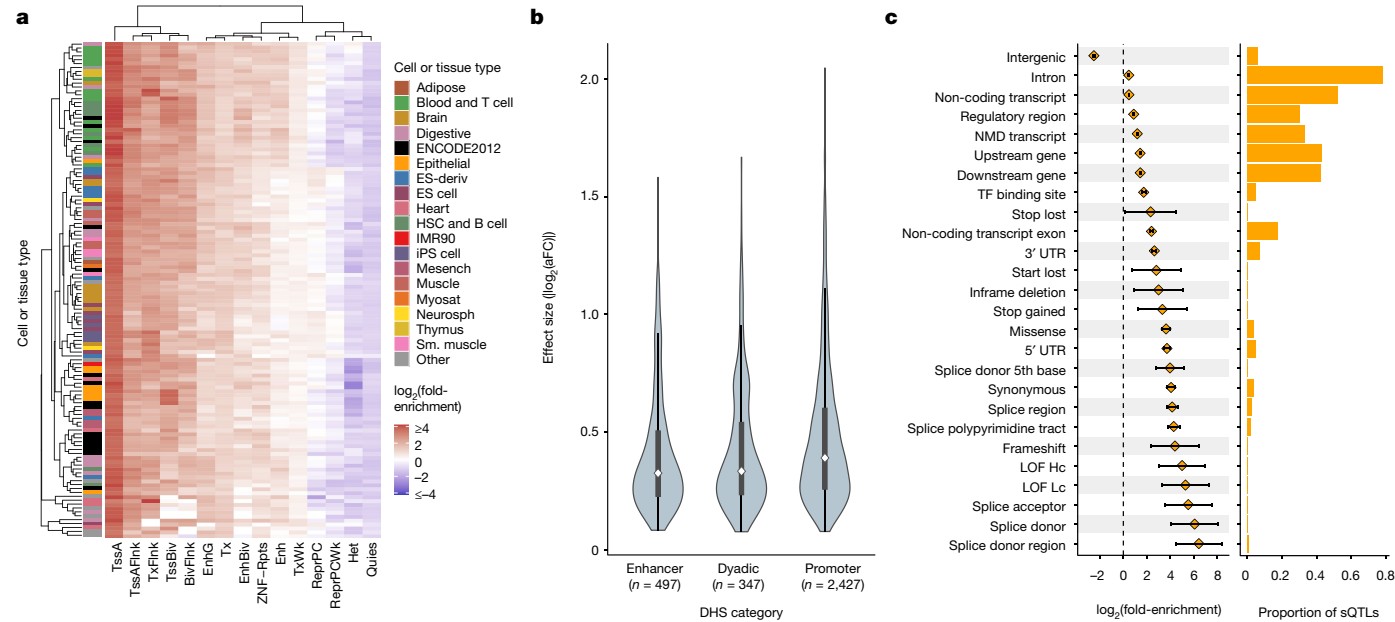

**Fig. 4 | Fine-mapped *cis*-QTLs are strongly enriched in regulatory regions across multiple cell and tissue types. a**, A heatmap representing hierarchical clustering of the enrichment of *cis*-eQTLs in predicted chromatin states using the Roadmap Epigenomics 15-state chromHMM model across 127 cell and tissue samples. **b**, Distribution of absolute value of lead *cis*-eQTL effect sizes measured as $\log_2$(aFC) across putatively active chromatin states of LCLs linked to multi-tissue DHSs. Sample sizes describe the number of unique eVariants annotated as belonging to each of the DHS categories. Bars represent the first, second (median) and third quartiles of the data and whiskers are bound to 1.5× the interquartile range. **c**, Enrichment of lead sQTLs ($n = 13,107$ unique sVariants total, at least 5 per category) within functional annotation categories from Ensembl Variant Effect Predictor (left), along with the proportion of all lead sQTLs falling into each annotation category (right). Error bars denote 95%

CI around the estimated sQTL enrichment in each category. Enrichment was calculated in comparison to a background set of variants matched on MAF and distance from the TSS. Annotation categories are not mutually exclusive and therefore sum to a proportion greater than 1. ES, embryonic stem; HSC, haematopoietic stem cell; iPS, induced pluripotent stem; Mesench, mesenchymal cell; Myosat, myosatellite cell; Neurosph, neurosphere; Sm., smooth; TssA, active TSS; TssAFlnk, flanking active TSS; TxFlnk, transcription at gene 5' and 3'; Tx, strong transcription; TxWk, weak transcription; EnhG, genic enhancer; Enh, enhancer; ZNF/Rpts, ZNF genes plus repeats; Het, heterochromatin; TssBiv, bivalent/poised TSS; BivFlnk, flanking bivalent TSS/enhancer; EnhBiv, bivalent enhancer; ReprPC, repressed polycomb; ReprPCWk, weak repressed polycomb; Quies, quiescent/low; NMD, nonsense-mediated mRNA decay; LOF, loss of function; Hc, high confidence; Lc, low confidence.

genome-wide association studies (GWAS). As a multi-ancestry resource, we anticipate that MAGE will facilitate the interpretation of GWAS from underrepresented populations. One such cohort is the Population Architecture using Genomics and Epidemiology (PAGE) study[8], which comprises 49,839 non-European individuals, including large samples of individuals who self-reported as Hispanic/Latin American or African American, as well as smaller samples of individuals who self-reported as Asian, Native Hawaiian or Native American. We performed colocalization analysis to identify shared signals between GWAS of 25 complex traits from PAGE and *cis*-eQTLs and *cis*-sQTLs from MAGE. PAGE GWAS data include quantitative biomedical traits such as platelet count and cholesterol levels, as well as diseases such as type 2 diabetes (see Supplementary Table 1 for a full list of the traits included in this analysis).

Across these 25 traits, we identified 384 independent GWAS signals. For each independent GWAS signal, we tested for eQTL colocalization with each eGene within 500 kbp. We implemented this analysis using a combination of SuSiE[25] and coloc[34,35] to allow for multiple causal variants at each signal and to allow for different patterns of LD between the two datasets. We defined moderate colocalizations as those with posterior probabilities ≥ 0.5 and strong colocalizations as those with posterior probabilities ≥ 0.8.

Using this approach, we identified moderate colocalizations with MAGE *cis*-eQTLs for 39 independent GWAS signals across 14 traits and strong colocalizations for 25 independent GWAS signals across 13 traits (Supplementary Fig. 18). These included 6 GWAS signals across 6 traits for which the GWAS variant was rare (minor allele frequency (MAF) < 0.05) or unobserved in the European continental group in the 1KGP. Among these, one notable result involved colocalization

($P_{coloc} = 0.998$) between a platelet count GWAS hit (sentinel variant rs73517714) and an eQTL hit of the tropomyosin gene *TPM4*, whereby the lead eQTL variant (rs143558304) falls within the 3' UTR. Previous work has implicated rare missense variants in *TPM4* with platelet abnormalities and excessive bleeding[36], findings that provide support for a role of this gene in platelet function. The MAGE lead eQTL and the GWAS sentinel variant were in strong LD ($R^2 = 0.874$ in MAGE) and were rare (MAF < 0.05) in the European continental group of the 1KGP but more common in the African continental group.

We repeated this colocalization analysis for MAGE sQTLs. Across the same set of 384 GWAS signals, we identified moderate colocalizations with MAGE *cis*-sQTLs for 30 independent GWAS signals across 12 traits and strong colocalizations for 24 independent GWAS signals across 10 traits (Supplementary Fig. 18). These included three GWAS signals across two traits for which the GWAS variant was rare or unobserved in the European continental group in the 1KGP. Together these results highlight the utility of paired globally diverse gene expression and WGS datasets like MAGE and 1KGP, respectively, in interpreting complex trait GWAS of non-European cohorts.

## Population-specificity of QTLs

A fundamental question in association studies is the extent to which genetic associations replicate across human groups and the underlying factors that drive heterogeneity between groups. Several previous studies have demonstrated that although QTL effects are strongly correlated across populations[12,37], the predictive power of association study summary statistics (for example, polygenic scores) declines

when applied to groups whose ancestry does not match that of the discovery sample[9,14]. The underlying causes of such poor portability is a topic of active debate[37,38], and several non-mutually exclusive explanations have been proposed: (1) differences in the allele frequency (AF) of causal variants between groups can lead to differential statistical power; (2) differences in patterns of LD (either between a tag variant and a causal variant or between multiple causal variants) between groups can lead to nominal effect size heterogeneity; and (3) epistatic interaction between multiple causal variants, one or both of which vary in AF across groups can lead to nominal effect size heterogeneity. Gene-by-environment interactions may also drive effect size heterogeneity, but we anticipate that such interactions are less relevant to our data given the common conditions used for deriving and culturing immortalized LCLs, as well as the block-randomized nature of the experimental design (Supplementary Methods).

To gain insight into the relative importances of these phenomena, we identified and characterized two broad classes of population-specific QTLs: (1) QTLs for which AF differs between continental groups (which we term frequency differentiated QTLs (fd-QTLs)) and (2) QTLs that exhibit effect size heterogeneity between continental groups (which we term heterogeneous effect QTLs (he-QTLs)). We consider each class in turn in the subsequent sections.

### Frequency differentiation of QTLs

We proposed that the diversity of our sample would facilitate discovery of new QTLs that are private to populations that were underrepresented in previous molecular association studies. To test this hypothesis, we evaluated the frequency distribution across continental groups of the 15,664 fine-mapped lead eQTLs in MAGE. We observed that 8,837 (56.4%) lead eQTLs are 'globally common' (MAF > 0.05 in each continental group), a result consistent with the fact that statistical power for eQTL discovery scales with MAF and that most common variation is shared across human populations[39,40] (Extended Data Fig. 4a,b). However, we also identified 1,310 (8.3%) lead eQTLs that are unobserved in the European continental group but present in one or more other continental groups (Extended Data Fig. 4c). An additional 115 (0.6%) lead eQTLs are unobserved in both European and African ancestry groups (Extended Data Fig. 4d). Qualitatively similar patterns were also apparent for sQTLs (Extended Data Fig. 5).

To further contextualize our results, we compared our eQTL fine-mapping data to that from GTEx, which largely comprises individuals of European ancestries and some African American individuals. To account for the multi-tissue nature of GTEx, we took the union of credible sets across tissues for a focal gene to compare with the credible sets for that same gene in MAGE (Supplementary Methods). Overall, we found that 8,069 MAGE credible sets (6,421 genes) replicated in GTEx compared with 7,595 credible sets (5,545 genes) that did not replicate (Fig. 5a). We additionally identified 701 genes with at least one credible set in MAGE but no apparent credible set in GTEx. Notably, we observed that lead eQTLs in MAGE that did not replicate in GTEx tended to exhibit greater geographical differentiation, with higher frequencies outside Europe relative to variants that replicated between studies, which tended to be common across all populations (Fig. 5a). Moreover, the 79,915 GTEx lead eQTLs that were not replicated in MAGE (7,913 lead eQTLs replicated) are enriched for tissue-specific effects (two-tailed Mann–Whitney $U$-test: $P < 10^{-10}$; Extended Data Fig. 6). This was despite showing qualitatively similar patterns of functional enrichment that support their biological validity (Supplementary Fig. 19). Together, these results highlight important aspects of experimental design across multiple axes of diversity, such as ancestry and tissue composition, that shape the statistical findings of molecular QTL studies.

One example of a fd-eQTL that we identified was rs115070172, for which the T allele was common (AF > 0.05) only within the Admixed American continental group in MAGE and was at high frequency

(AF = 0.63) in the Peruvian population (Fig. 5b). This variant was the lead eQTL for one of two credible sets of *GSTP1*, a tumour suppressor gene for which its expression has been implicated in breast cancer[41–43]. The T allele of the rs115070172 variant was significantly associated with lower expression of *GSTP1* (Fig. 5c). Intersection with epigenomic data indicated that this fine-mapped lead eQTL lies within a putative enhancer region (Supplementary Fig. 20). Notably, the expression of *GSTP1* was significantly lower in individuals from the Peruvian population compared with other global populations (Fig. 5d), and we propose that this eQTL signal may explain this trend.

To more broadly examine the role of fd-eQTLs in driving differential gene expression between continental groups, we quantified $F_{ST}$ values[44] for each lead eQTL and intersected these values with the differential expression results for the respective eGene for each eQTL (Fig. 5e). Among continental groups, differentially expressed eGenes (FDR-adjusted $P \leq 0.05$) possessed higher $F_{ST}$ values than non-differentially expressed eGenes (two-tailed Mann–Whitney $U$-test: $Z = 0.022 \pm 0.001$, 95% CI; $P < 0.05$). This result suggests that gene expression differences across populations can be attributed to frequency differentiation of causal eQTLs.

### Consistency of eQTL effects

We sought to test for he-eQTLs in MAGE given recent debates about their prevalence and causes[8,15,37,38,45]. Because the genotypes are derived from high-coverage whole-genome sequencing in the 1KGP, MAGE should be robust to effect size heterogeneity resulting from population-specific LD patterns with untyped casual variants (which commonly affects microarray data), barring large structural variation that may escape detection with short-read sequencing. This enables investigation of other sources of effect size heterogeneity.

For each fine-mapped eGene, we first assessed whether its top nominal pass eQTL exhibited effect size heterogeneity between continental groups by fitting a model that included a genotype-by-continental group interaction term. Across 8,376 top nominal pass eQTLs that passed filtering (MAF ≥ 0.05 in at least two continental groups), 70 (0.84%) exhibited a significant genotype-by-continental group interaction after Bonferroni correction (Fig. 5f). Notably, we observed that eGenes with more fine-mapped credible sets were more likely to exhibit significant interaction effects, which suggested that the additive effects of multiple causal variants may drive apparent interaction effects.

To test this hypothesis, we discovered he-eQTLs from among our fine-mapped eQTLs. For each fine-mapped eGene, we included the lead eQTL from each of its credible set (or sets) as predictors and tested for a genotype-by-continental group interaction effect for one lead eQTL at a time. Supporting our hypothesis, 64 (91%) of eGenes with a significant interaction effect had no significant interaction effects after controlling for the additive effects of multiple causal signals for that gene (Fig. 5f). The few remaining interaction effects (9 eQTLs; 0.07% of all eQTLs that passed filtering) may be driven by non-additive epistatic interactions between variants, by additional untested causal variants that did not meet nominal MAF thresholds or by population-specific LD patterns with untyped causal variants. Qualitatively similar patterns were observed when testing for interactions between genotype and global genotype principal components (Extended Data Fig. 7).

An alternative approach based on stratified eQTL mapping and fine mapping within each continental group (Supplementary Methods) likewise indicated high consistency in effect sizes, such that 97.5–99.8% of credible sets had similar effect sizes between pairs of continental groups (Extended Data Fig. 8 and Supplementary Fig. 21). Together, these results indicate that effect size heterogeneity of eQTLs between populations is rare, and apparent heterogeneity may instead reflect the failure to control for the additive effects of multiple independent causal signals.

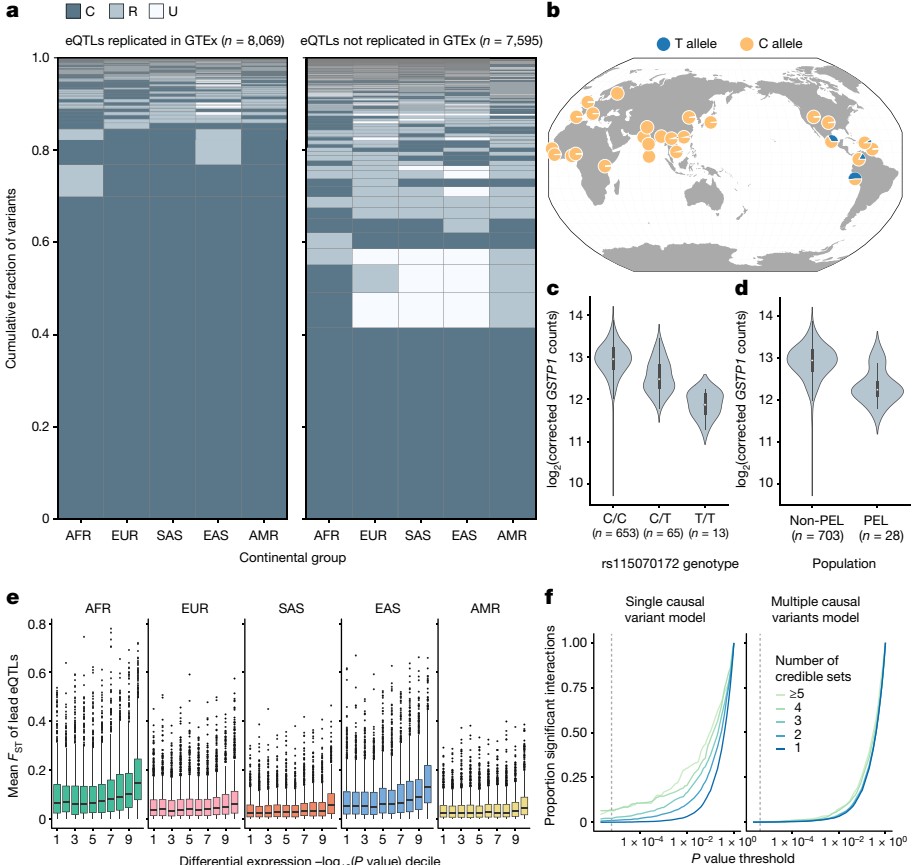

**Fig. 5 | Population-specific genetic effects on gene expression. a**, Joint distribution of AFs of fine-mapped lead eQTLs across continental groups in the 1KGP[6], stratifying on replication status in GTEx[26]. Variants are categorized as unobserved (U; MAF = 0), rare (R; MAF < 0.05) or common (C; MAF ≥ 0.05) within each continental group. **b**, AF of a lead eQTL of *GSTP1* (rs115070172) across populations from the 1KGP. **c**, Expression of *GSTP1*, stratified by genotype of rs115070172. Sample sizes describe the number of MAGE samples with each genotype. **d**, Expression of *GSTP1*, stratified by population label (PEL versus non-PEL). Sample sizes describe the number of MAGE samples in each population category. **e**, Mean $F_{ST}$ value between the focal continental group and all other groups of lead eQTLs, stratifying by differential expression decile (contrasting the focal continental group with all other groups) of respective eGenes, where the 10th decile represents the most significant differential expression (n = 9,807

genes total). Differential expression *P* values (two-tailed) were obtained for each gene using a negative binomial generalized linear model contrasting each continental group with all other samples. For **c**–**e**, bars represent the first, second (median) and third quartiles of the data and whiskers are bound to 1.5× the interquartile range. For **e**, data outside whisker ranges are shown as dots. **f**, Number of significant genotype-by-continental group interactions at varying *P* value thresholds (dashed line denotes Bonferroni threshold) for a model that considers a single causal variant (left) versus multiple potential causal variants per gene (right), stratifying by the number of credible sets for the gene. *P* values are one-tailed and are obtained from a *F*-test. Map in **b** reproduced using the browser described in ref. 59, under a Creative Commons licence CC BY 4.0.

## Discussion

Combined with existing whole-genome sequencing data from the same samples[24], MAGE offers a large open-access dataset for studying the diversity and evolution of human gene expression and splicing. Our study also offers insight into the genetic sources of variation in these key molecular phenotypes, which may in turn mediate variation in organismal traits. By evenly spanning samples from all 26 populations of the 1KGP[6], MAGE includes several ancestry groups that were poorly represented in previous molecular association studies[7].

The scale and diversity of the dataset enabled the discovery of numerous potentially new genetic associations while offering high resolution for identifying putatively causal variants and elucidating their mechanisms of action. Our study also demonstrated that the effect sizes of eQTLs are highly consistent across populations, which implies that *trans*-genetic effects (driven by global ancestry patterns), if adequately controlled for, generally do not have a strong impact on the effects of causal variants in *cis*. Although addressing a point of recent debate within the field[8,15,37,38,45,46], this conclusion

is in agreement with several previous studies that used orthogonal approaches for evaluating effect size heterogeneity based on analysis of admixed individuals[37,45]. This finding is encouraging for predictive applications such as polygenic risk scores and transcriptome-wide association studies, as it suggests that models that focus on causal signals and do not make assumptions about the number of such signals may exhibit better portability between groups. The extent to which this conclusion applies to more polygenic complex traits is an open question, but a recent study that investigated effect size heterogeneity in admixed individuals across 38 complex traits found that nominal effect sizes are consistent across local ancestries[37]. Such consistency of genetic effects further motivates the use of diverse samples for association studies, as a common causal variant identified in one population may inform the effect of that variant in a population in which the same variant is rare and association testing would be underpowered. Thus, all populations—not only underrepresented populations—benefit from the inclusion of greater diversity in genetic studies, which empowers more accurate and generalizable predictions for personalized medicine[47].

Intersection of eQTLs and sQTLs with data from GWAS may facilitate understanding of the molecular mechanisms that link genetic variation to organismal phenotypes. Using GWAS data from the PAGE study of ancestrally diverse individuals, we identified 54 GWAS signals that colocalize with eQTL and sQTL signals. Although informative and substantial in absolute number, these reflect a minority of all GWAS hits. Limited colocalization between molecular QTLs and GWAS hits is well described, largely stemming from distinct selective pressures shaping genetic variation that can be identified (with incomplete statistical power) in the two analyses[48]. GWAS hits tend to occur within genes under strong purifying selection, whereas molecular QTLs are most easily identified for genes under relaxed constraint. This is consistent with our finding that genes exhibiting strong signals of selection are depleted of MAGE eQTLs. Although inclusion of additional tissues and cell types modestly increases the rate of colocalization, these qualitative observations hold even for multi-tissue studies such as GTEx[48]. Despite these general limitations of colocalization analyses, our results demonstrated instances in which MAGE facilitated the interpretation of GWAS results, particularly in underrepresented populations. We anticipate that this utility will further improve as GWAS continue to expand to more diverse cohorts.

By design, our study focused on a single cell type of LCLs, which offers a useful model for studying gene expression given their low somatic mutation rates and robust gene expression patterns encompassing key metabolic pathways[49]. Although this enabled us to mitigate the effects of environmental variation and to compare our results to related studies performed in the same cell lines[4], future studies may seek to understand ancestry differences in expression across developmental, cellular and other environmental contexts, including with respect to dynamic QTLs for which effects vary based on those contexts[50]. Future studies of diverse cohorts may also leverage new technologies (such as long-read genome, cDNA or direct RNA-seq[51–53]) to achieve higher resolution for isoform detection as well as improved analysis of genes that occur within highly repetitive or structurally complex regions.

Finally, although geographically diverse, the sampling of the 1KGP is not without biases, for example, narrowly sampling the vast diversity within Africa and excluding indigenous populations from Oceania and the Americas, as well as countless other populations. Addressing these biases will require deeper community engagement and respect for the rights, interests and expectations of research participants from diverse human groups[54]. This expansion of diversity in functional genomics parallels efforts for improved representation of diversity in genome sequencing and assembly, including construction of pangenomes[55,56]. Although the current study was based on alignment to a linear representation of the reference genome, given the maturity of software tools and annotations built on this paradigm, MAGE offers a valuable data resource for testing pangenomic methods over the coming decade as they are developed by the research community.

Our work provides a more complete picture of the links between genetic variation and genome function across diverse populations, as well as the evolutionary forces that have shaped this variation within our species. Complemented by existing high-coverage whole-genome sequencing data, we anticipate that this dataset will serve as a valuable resource to facilitate future research into the complex genetic basis of variation in human genome function.

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

## Reporting summary

Further information on research design is available in the Nature Portfolio Reporting Summary linked to this article.

## Data availability

Newly generated RNA-seq data for the 731 individuals (779 total libraries) are available from the Sequence Read Archive (accession: PRJNA851328). Processed gene expression matrices and QTL mapping results are available from Zenodo (https://doi.org/10.5281/zenodo.10535719)[60]. Alignment was performed using the GRCh38 reference genome (https://www.ncbi.nlm.nih.gov/datasets/genome/GCA_000001405.15/), and gene and transcript annotations were obtained from GENCODE (v.38; https://www.gencodegenes.org/human/release_38.html). Our study also relied on published whole-genome sequencing data from 1KGP samples (https://www.internationalgenome.org/data-portal/data-collection/30x-grch38). QTL mapping results were compared with GTEx (dbGaP accession: phs000424.v9.p2) and Geuvadis (EBI Array-Express accessions: E-GEUV-1, E-GEUV-2 and E-GEUV−3). Functional enrichment analyses relied on annotations from ENCODE (https://www.encodeproject.org/data/annotations/) and Roadmap Epigenomics (https://egg2.wustl.edu/roadmap/web_portal/chr_state_learning.html). Colocalization analyses were conducted using data from the PAGE study (harmonized GWAS sumstats from the GWAS Catalog: https://www.ebi.ac.uk/gwas/publications/31217584; genotypes from TOPMED, dbGaP accession: phs001974.v5.p1).

## Code availability

Code used for the analyses presented in this paper is available on GitHub (https://github.com/mccoy-lab/MAGE) and archived on Zenodo (https://doi.org/10.5281/zenodo.10072080)[61].

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

**Acknowledgements** Thanks to S. Montgomery and members of the Montgomery Laboratory; J. Leek; members of the McCoy Lab, the Department of Biology and the Center for Computational Biology at Johns Hopkins for helpful discussion and feedback; C. Runnels for initial exploration of signatures of selection; B. Huang for initial exploration of splicing enrichments; the staff of Advanced Research Computing at Hopkins for support; and M. Mitchell and other staff at the Coriell Institute, P. Boyle, B. Kerr, A. James, A. Bronzato-Badial, M. Carter and other staff at Genewiz/Azenta for assistance in generating RNA-seq data. R.C.M., M.G.T. and A. Biddanda are supported by NIH/NIGMS award R35GM133747 and NIH Award OT2OD034190. D.J.T. is supported by NIH/NHGRI award F31HG012900. S.M.Y. is supported by NIH/NHGRI award F31HG012495. A. Battle is supported by NIH/NIGMS award R35GM139580. G.L.W. is supported by NIH/NHGRI award R35HG011944. The content is solely the responsibility of the authors and does not necessarily represent the official views of the National Institutes of Health.

**Author contributions** D.J.T. and R.C.M. initially conceived the project. Raw data generation was managed by D.J.T. and R.C.M. Alignment was done by R.C.M. Expression quantification was done by D.J.T. and M.G.T. Splicing quantification was done by M.G.T. PCA and ADMIXTURE comparisons with previous datasets were done by R.C.M. Analyses of between and within population expression and splicing diversity were done by D.J.T. and M.G.T. All eQTL and sQTL mapping and fine mapping, as well as eQTL effect size estimation were done by D.J.T. Intersection of eQTLs with selection metrics was done by R.C.M. and A. Biddanda. Functional annotation and characterization of eQTLs and sQTLs and subsequent analyses were performed by S.B.C. and A. Battle. Colocalization analysis with PAGE was performed by D.J.T. with data and guidance provided by G.L.W. Quantification of population-stratified eQTLs and sQTLs (fd-QTLs) was performed by A. Biddanda. Comparison of MAGE and GTEx fine-mapped eQTLs was performed by A. Biddanda. Analyses of the correlation between fd-eQTLs and differential gene expression between populations was performed by M.G.T. Interaction test-based discovery of he-eQTLs was performed by D.J.T. Population-stratified discovery of eQTLs and subsequent he-eQTL analysis was performed by S.M.Y. D.J.T. curated published output data. D.J.T., S.B.C., M.G.T., A. Biddanda, S.M.Y. and R.C.M. wrote the manuscript. D.J.T. and R.C.M. edited the manuscript, with the assistance of all authors.

**Competing interests** A. Battle is a co-founder and equity holder of CellCipher, a stockholder in Alphabet, and has consulted for Third Rock Ventures. The other authors declare no competing interests.

### Additional information

**Correspondence and requests for materials** should be addressed to Rajiv C. McCoy.

**A.**

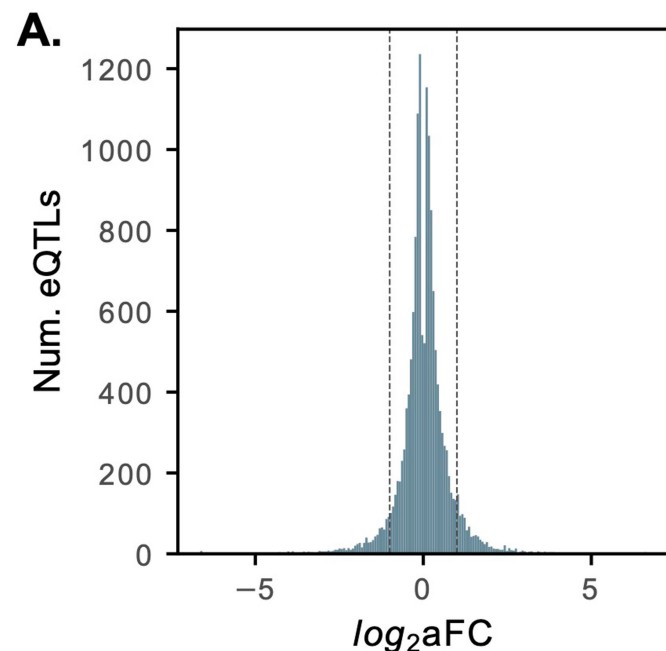

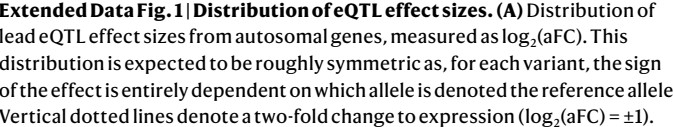

**B.**

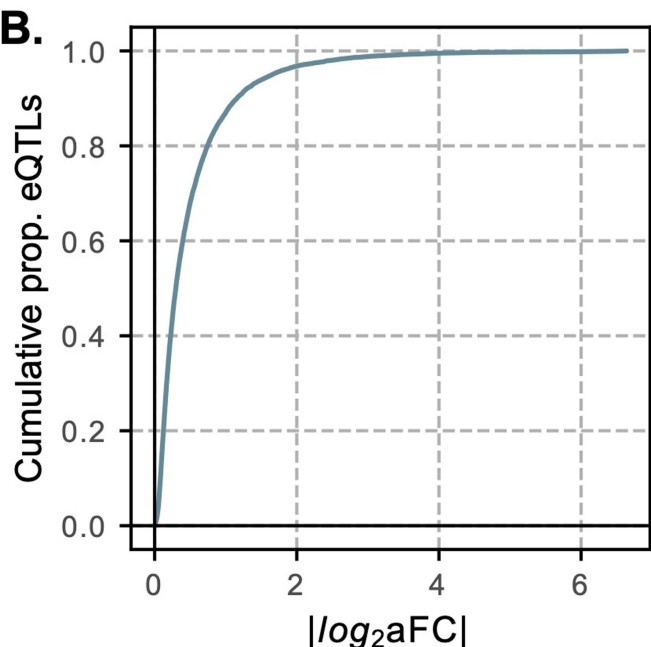

**Extended Data Fig. 1 | Distribution of eQTL effect sizes. (A)** Distribution of lead eQTL effect sizes from autosomal genes, measured as $\log_2$(aFC). This distribution is expected to be roughly symmetric as, for each variant, the sign of the effect is entirely dependent on which allele is denoted the reference allele. Vertical dotted lines denote a two-fold change to expression ($\log_2$(aFC) = ±1).

Most eQTLs have a relatively small effect on expression level. **(B)** Cumulative distribution of the absolute value of effect size across lead eQTLs from autosomal genes. Only 2031 (13%) lead eQTLs had greater than a twofold effect on gene expression (median $|\log_2$(aFC)$| = 0.30$).

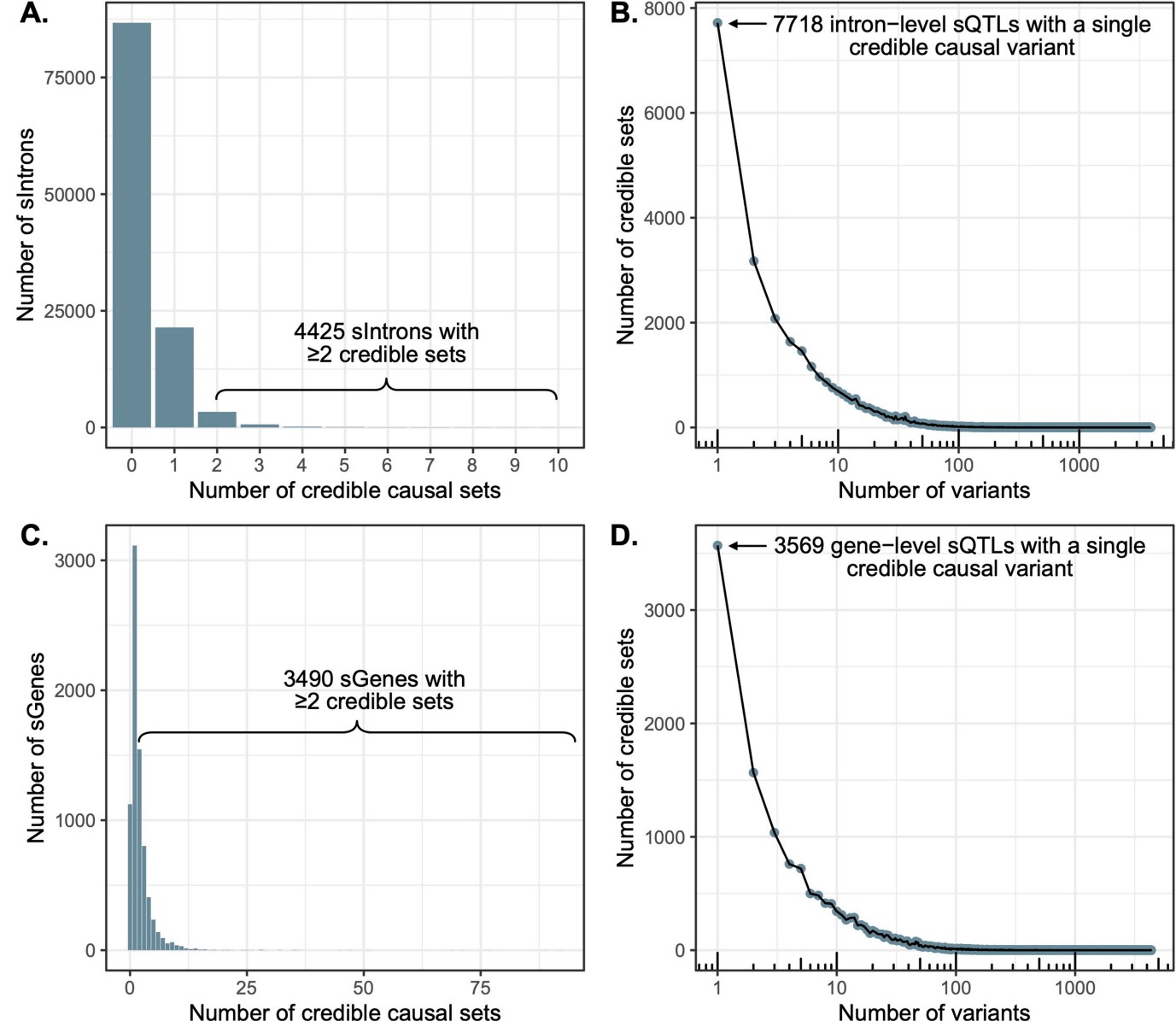

**Extended Data Fig. 2 | Mapping of high-resolution sQTLs. (A)** Number of credible sets per sIntron, where we define sIntrons as all introns (that passed filtering) for autosomal genes identified as sGenes in the FastQTL permutation pass. We ran SuSiE separately for each sIntron. **(B)** Resolution of sIntron fine-mapping, defined as the number of variants per credible set. **(C)** After fine-mapping, overlapping intron-level credible sets were iteratively merged to produce gene-level credible sets. Panel C shows the number of merged credible sets per sGene. **(D)** The resolution of sGene fine-mapping, defined as the number of variants per merged credible set. These results demonstrate evidence of widespread allelic heterogeneity whereby multiple causal variants independently modulate splicing patterns of the same genes.

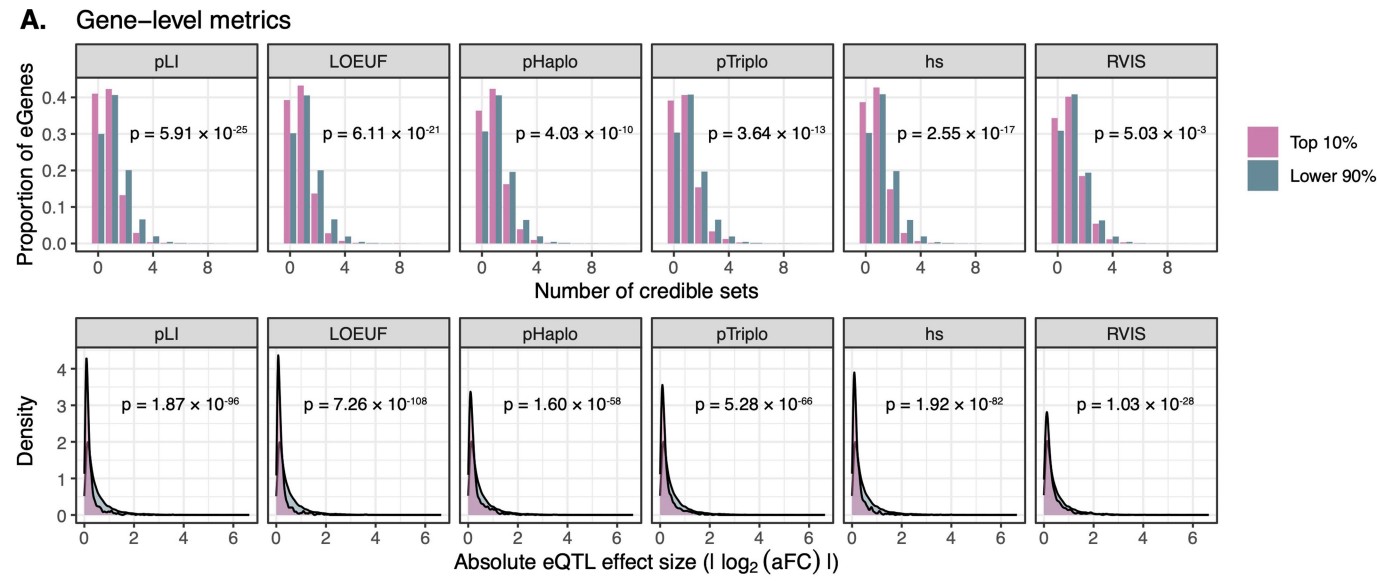

**A. Gene–level metrics**

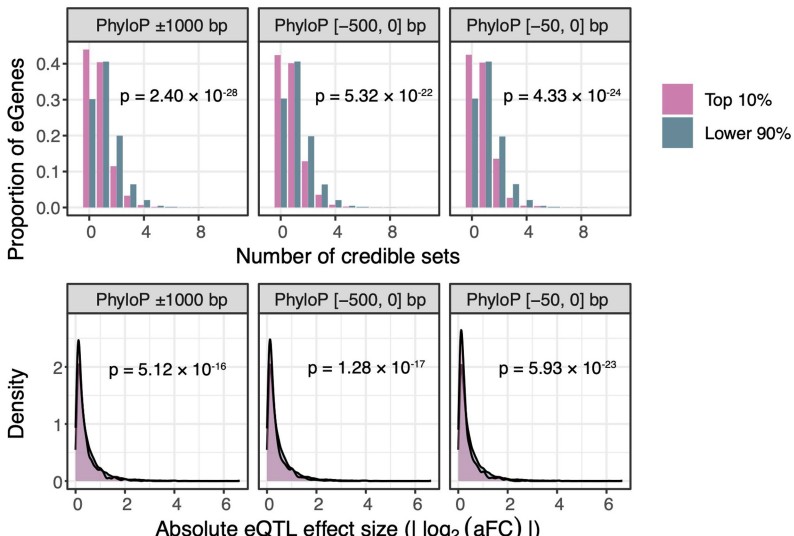

**B. Promoter–level metrics**

**Extended Data Fig. 3 | Evidence of negative selection on expression-altering variation across a range of mutational constraint metrics.** (A) Top row: number of credible causal sets for genes in (pink) and outside (blue) the top decile of various gene-level constraint metrics (pLI, LOEUF, pHaplo, pTriplo, hs, RVIS) obtained from the literature. P-values are two-tailed and based on quasi-Poisson generalized linear models that include mean expression level as a covariate. Bottom row: effect sizes (|log2(aFC)|) of lead eQTLs within (pink) and outside (blue) the same categories. P-values are two-tailed and based on Wilcoxon rank sum tests. (B) Same as panel A, but for mean PhyloP scores summarizing conservation among genome sequence alignments of 447 mammals within putative promoter elements, defined based on intervals around the TSS ([−1000, 1000] bp, [−500, 0] bp, [−50, 0] bp).

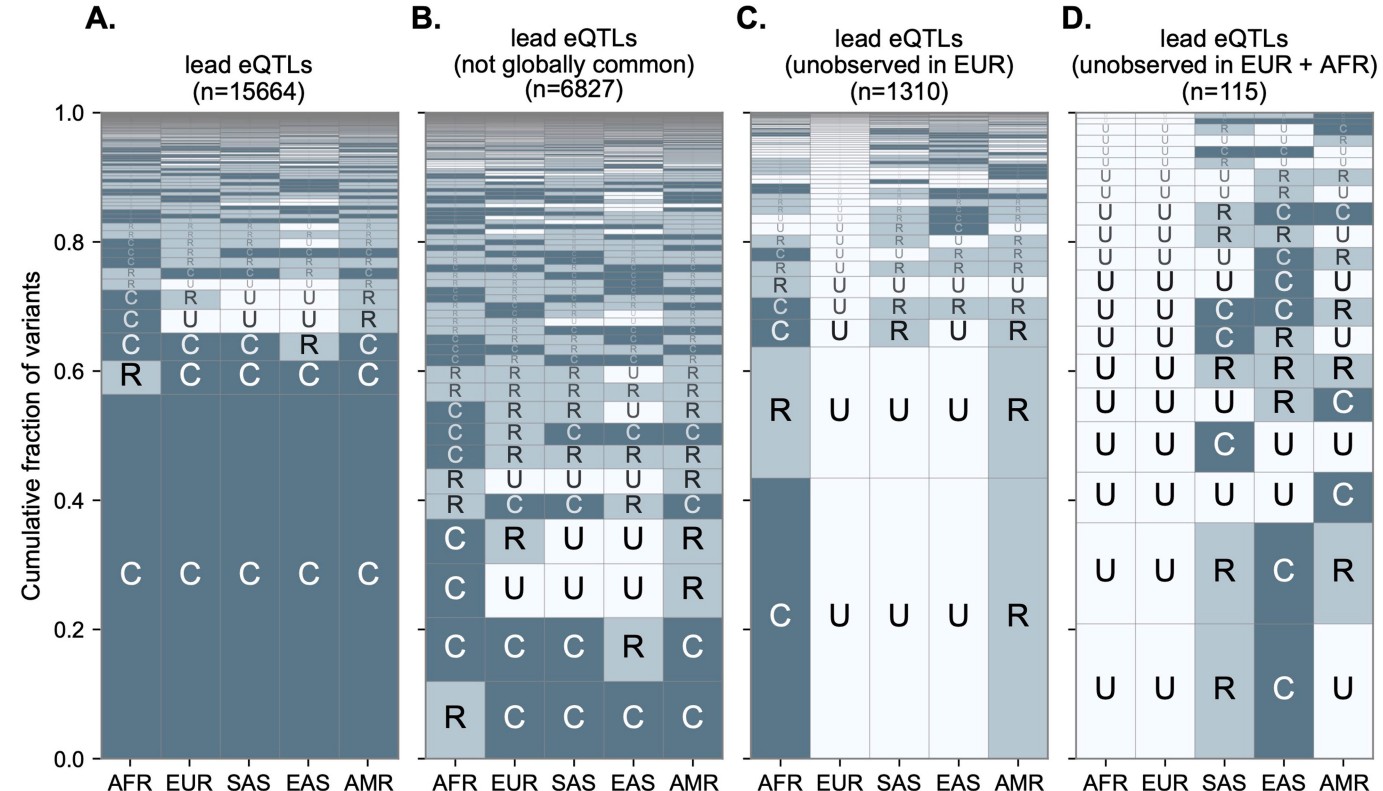

**Extended Data Fig. 4 | Population stratification of eQTLs.** Geographic frequencies of autosomal lead eQTLs found in MAGE across **(A)** all lead eQTLs, **(B)** excluding variants with allele frequencies > 5% across all continental groups, **(C)** only including variants unobserved in the European continental group, and **(D)** only including variants unobserved in both the European and African continental groups. The geographic distributions are sorted with the most common at the bottom and rarest at the top. Allele frequencies are categorized as unobserved (U), rare variants with allele frequencies <5% (R), and common variants (C) with allele frequencies ≥5%.

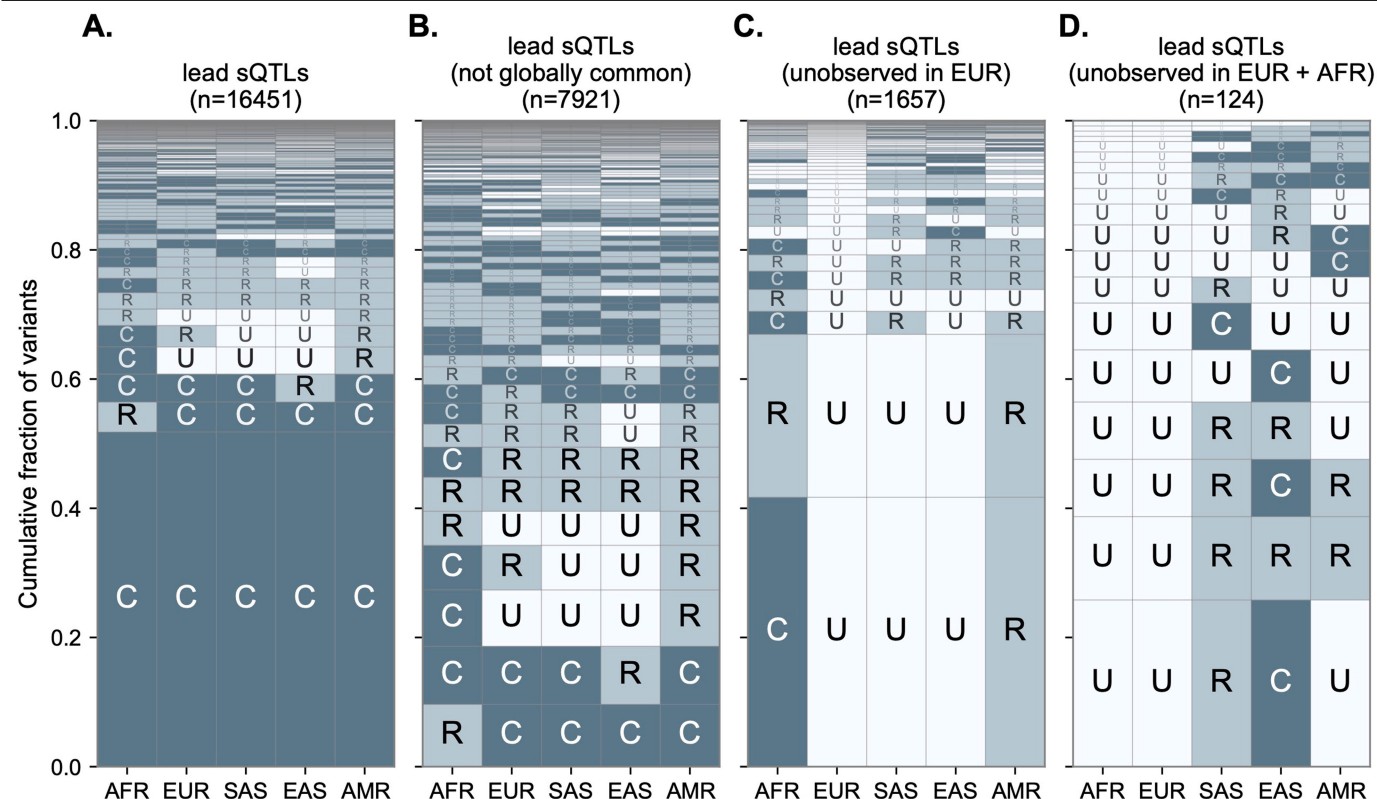

**Extended Data Fig. 5 | Population stratification of sQTLs.** Geographic frequencies of autosomal lead sQTLs found in MAGE across **(A)** all lead sQTLs, **(B)** excluding variants with allele frequency > 5% across all continental groups, **(C)** only including variants unobserved in the European continental group, and **(D)** only including variants unobserved in both the European and African continental groups. Allele frequencies are categorized as unobserved (U), rare variants with allele frequencies <5% (R), and common variants (C) with allele frequencies ≥5%.

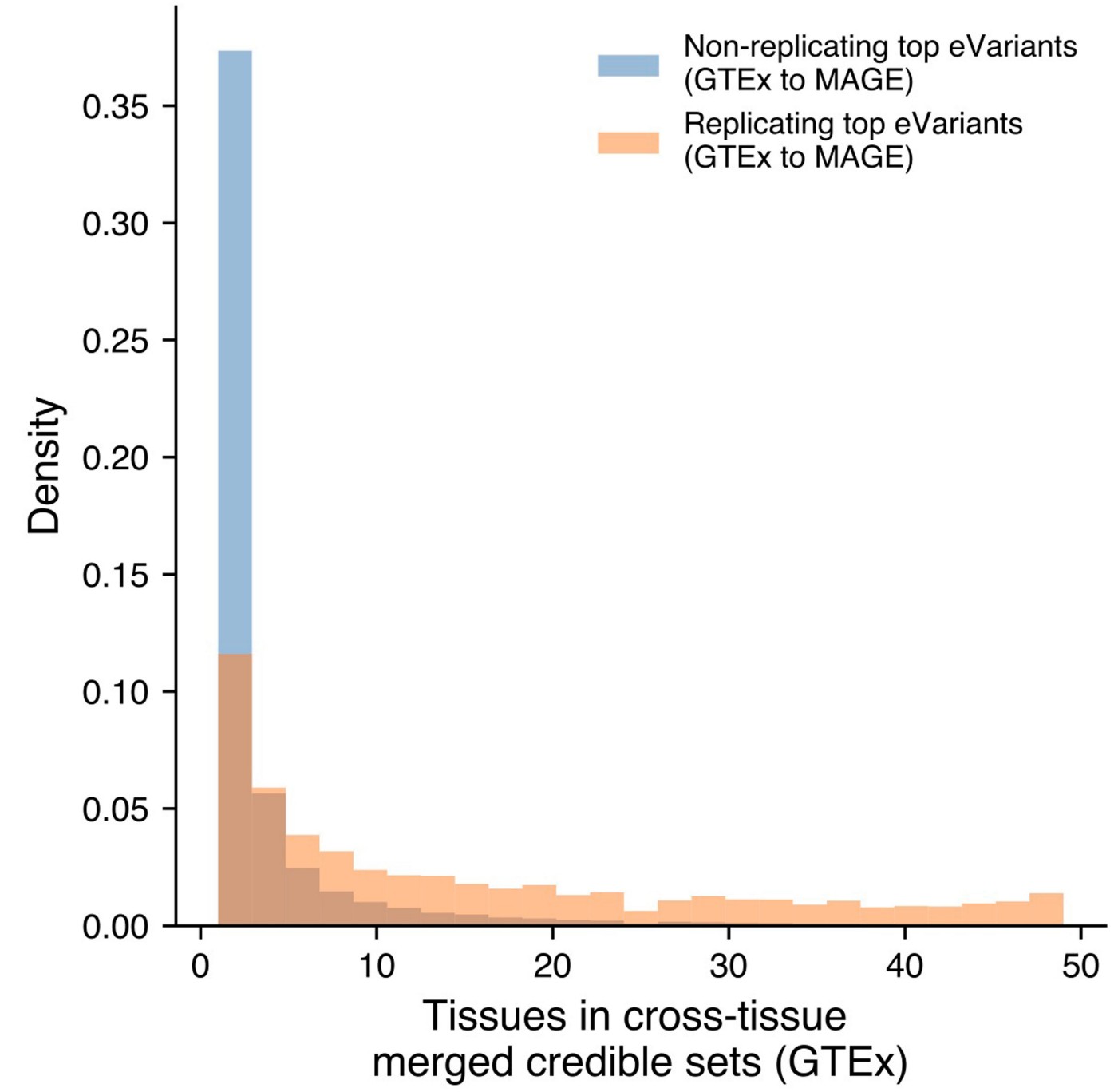

**Extended Data Fig. 6 | GTEx DAP-G fine-mapping signals that do not replicate in MAGE are largely tissue-specific.** Comparison of number of tissues contained by 79,915 cross-tissue merged credible sets (ctmCS) from GTEx that do not replicate in MAGE against 7,913 ctmCS that replicate in MAGE. The number of tissues is defined as the number of tissues across all variants included in a ctmCS.

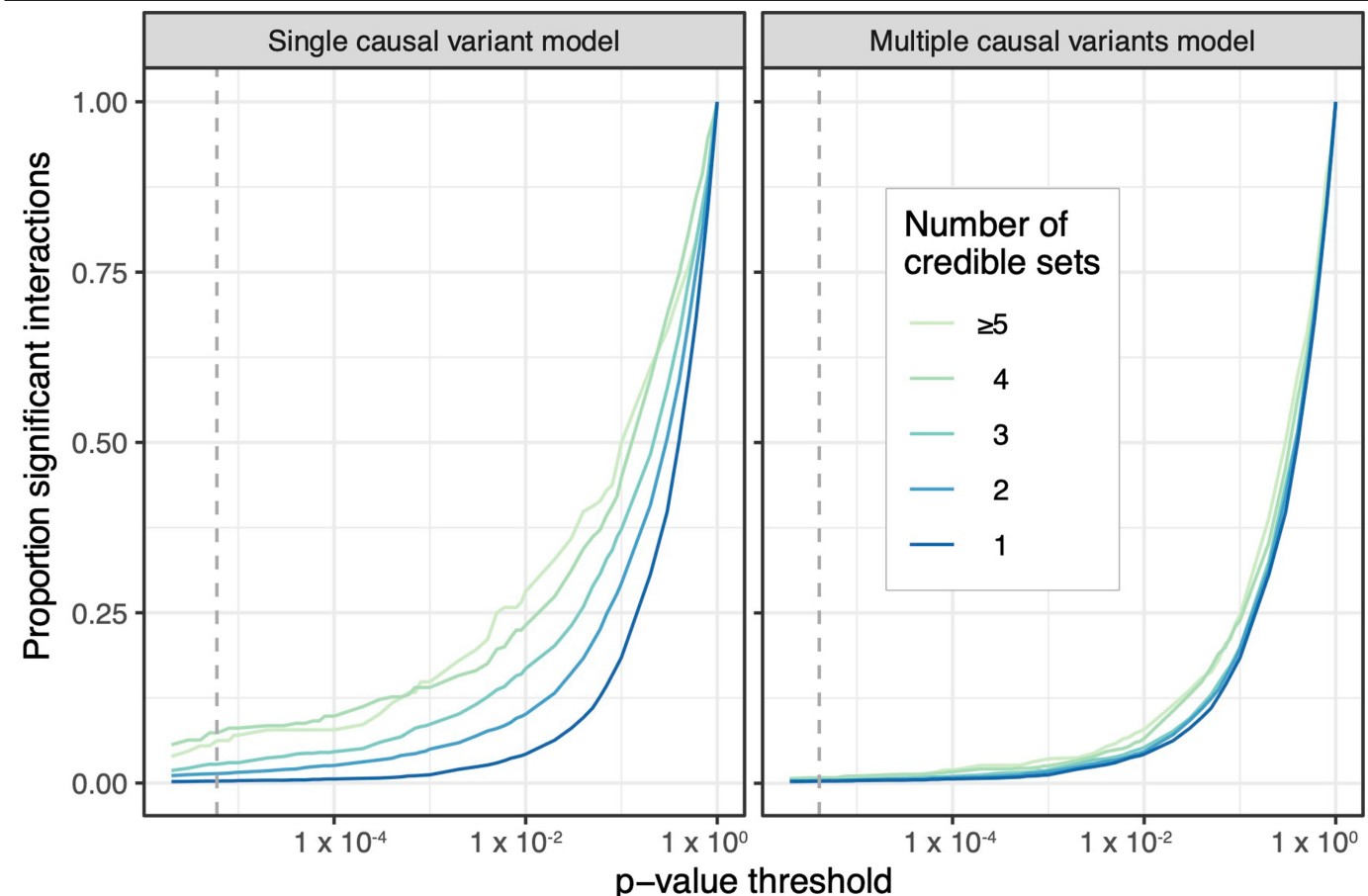

**Extended Data Fig. 7 | Results of eQTL genotype-by-principal component interaction test.** Analogous to Fig. 5f, the number of significant genotype-by-principal component interactions at varying p-value thresholds (dashed line denotes Bonferroni threshold) for a model that considers a single causal variant for each gene (left panel) versus a model that jointly considers multiple potential causal variants per gene (right panel). Genotype-by-principal component interactions were included for each of the top 5 global genotype principal components. Results are stratified by the number of credible sets for the gene (from one to five or greater). As in Fig. 5f, p-values are one-tailed and are obtained from an F-test comparing a model with an interaction term to one without.

**A.**

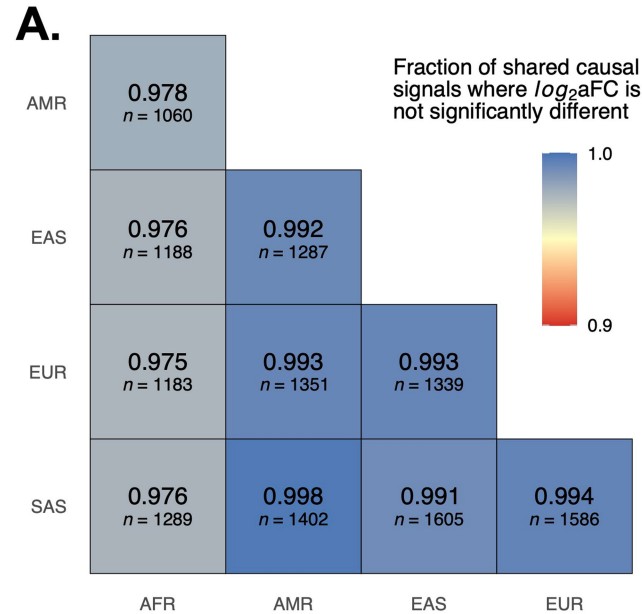

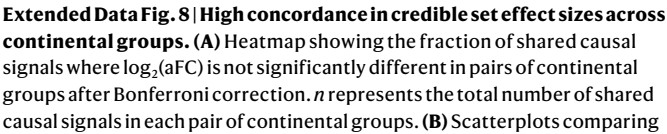

**B.**

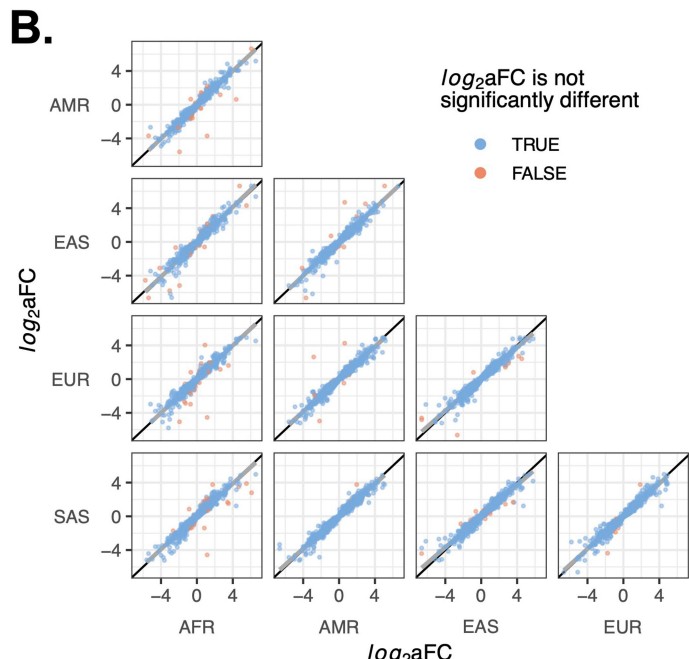

**Extended Data Fig. 8 | High concordance in credible set effect sizes across continental groups. (A)** Heatmap showing the fraction of shared causal signals where $log_2$(aFC) is not significantly different in pairs of continental groups after Bonferroni correction. $n$ represents the total number of shared causal signals in each pair of continental groups. **(B)** Scatterplots comparing $log_2$(aFC) within pairs of continental groups. Points are colored by whether the effect sizes are significantly different. The black line plots y = x (i.e., theoretical identical effect sizes). The gray line plots the best fit linear trendline. Significance is based on Bonferroni-corrected p-values (two-tailed) from a Welch's t-test of $log_2$(aFC) values in each pair of continental groups.

# Reporting Summary

## Statistics

For all statistical analyses, confirm that the following items are present in the figure legend, table legend, main text, or Methods section.

| n/a | Confirmed | |
|---|---|---|
| ☐ | ☒ | The exact sample size (*n*) for each experimental group/condition, given as a discrete number and unit of measurement |
| ☐ | ☒ | A statement on whether measurements were taken from distinct samples or whether the same sample was measured repeatedly |
| ☐ | ☒ | The statistical test(s) used AND whether they are one- or two-sided<br>*Only common tests should be described solely by name; describe more complex techniques in the Methods section.* |
| ☐ | ☒ | A description of all covariates tested |
| ☐ | ☒ | A description of any assumptions or corrections, such as tests of normality and adjustment for multiple comparisons |
| ☐ | ☒ | A full description of the statistical parameters including central tendency (e.g. means) or other basic estimates (e.g. regression coefficient) AND variation (e.g. standard deviation) or associated estimates of uncertainty (e.g. confidence intervals) |
| ☐ | ☒ | For null hypothesis testing, the test statistic (e.g. *F*, *t*, *r*) with confidence intervals, effect sizes, degrees of freedom and *P* value noted<br>*Give P values as exact values whenever suitable.* |
| ☒ | ☐ | For Bayesian analysis, information on the choice of priors and Markov chain Monte Carlo settings |
| ☒ | ☐ | For hierarchical and complex designs, identification of the appropriate level for tests and full reporting of outcomes |
| ☐ | ☒ | Estimates of effect sizes (e.g. Cohen's *d*, Pearson's *r*), indicating how they were calculated |

*Our web collection on statistics for biologists contains articles on many of the points above.*

## Software and code

Policy information about availability of computer code

| | |
|---|---|
| Data collection | No software was used for data collection. |
| Data analysis | All scripts and commands used to prepare and analyze the data can be found in a public GitHub repository (https://github.com/mccoy-lab/MAGE) and stable Zenodo repository (doi: 10.5281/zenodo.10072080) and are described in the Methods section.<br><br>The data analysis packages/tools used in this manuscript are as follows:<br>* ADMIXTURE tool (version 1.3.0)<br>* Plink tool (version 1.90b6.21)<br>* Salmon tool (version 1.5.2)<br>* tximport R package (version 1.18.0 and 1.24.0)<br>* STAR tool (version 2.7.10a)<br>* Leafcutter tool (version 0.2.9)<br>* regtools tool (version 0.5.2)<br>* MANTA R package (version 1.0.0)<br>* car R package (verzion 3.1-2)<br>* DESeq2 R package (version 1.36.0)<br>* stats R package (version 4.3.0)<br>* lme4 R package (version 1.1-34)<br>* EdgeR R package (version 3.32.1)<br>* peertool tool (version 1.0)<br>* FastQTL tool (version 2.184_gtex) |

For manuscripts utilizing custom algorithms or software that are central to the research but not yet described in published literature, software must be made available to editors and reviewers. We strongly encourage code deposition in a community repository (e.g. GitHub). See our web collection on software and code for full details.

```
* susieR R package (version 0.12.16)
* aFC-n tool (version 1.0.0 modified in-house, available here: https://github.com/dtaylo95/aFCn)
* GenomicRanges R package (version 1.38.0)
* bedtools tool (version 2.29.2)
* GREGOR tool (version 1.3.1)
* Ensembl Variant Effect Predictor (VEP) tool (version 109)
* LOFTEE plug-in for VEP (version 1.0.2)
* coloc R package (version 5.2.3)
* geovar python package (version 1.0.2)
* vcflib tool (version 1.0.0_rc2)
* statsmodels python package (version 0.14.0)
```

For manuscripts utilizing custom algorithms or software that are central to the research but not yet described in published literature, software must be made available to editors and reviewers. We strongly encourage code deposition in a community repository (e.g. GitHub). See the Nature Portfolio guidelines for submitting code & software for further information.

# Data

Policy information about availability of data

All manuscripts must include a data availability statement. This statement should provide the following information, where applicable:

- Accession codes, unique identifiers, or web links for publicly available datasets
- A description of any restrictions on data availability
- For clinical datasets or third party data, please ensure that the statement adheres to our policy

All sequencing data is available from the NCBI Sequence Read Archive (Accession: PRJNA851328). Processed gene expression matrices and QTL mapping results are available on Zenodo (doi: 10.5281/zenodo.10535719).

External datasets used:
* GRCh38 reference genome (https://www.ncbi.nlm.nih.gov/datasets/genome/GCA_000001405.15/)
* Sample genotypes from the New York Genome Center (NYGC) high-coverage WGS of the 1000 Genomes Project (1KGP; https://ftp.1000genomes.ebi.ac.uk/vol1/ftp/data_collections/1000G_2504_high_coverage/working/20201028_3202_phased/; https://www.internationalgenome.org/data-portal/data-collection/30x-grch38)
* Sex, continental group, and population labels for 1KGP samples (filtered to 30x GRCh38 samples; https://www.internationalgenome.org/data-portal/sample)
* Sample genotypes for GTEx v9 (dbGaP accession phs000424.v9.p2; https://www.ncbi.nlm.nih.gov/projects/gap/cgi-bin/study.cgi?study_id=phs000424.v9.p2)
* Transcript sequences and gene annotations from GENCODE v38 (https://www.gencodegenes.org/human/release_38.html)
* pLI and LOEUF (https://gnomad.broadinstitute.org/downloads#v4-constraint)
* pHaplo and pTriplo (https://zenodo.org/records/6347673)
* hs (https://github.com/agarwal-i/loss-of-function-fitness-effects)
* RVIS (https://doi.org/10.1371/journal.pgen.1003709.s002)
* PhyloP (https://hgdownload.soe.ucsc.edu/goldenPath/hg38/phyloP447way/)
* ENCODE TF binding sites (https://genome.ucsc.edu/cgi-bin/hgTrackUi?db=hg38&g=encRegTfbsClustered)
* Roadmap Epigenomics chromHMM and DHS annotations (https://egg2.wustl.edu/roadmap/web_portal/chr_state_learning.html)
* GWAS Catalog harmonized GWAS summary statistics from the PAGE study (https://www.ebi.ac.uk/gwas/publications/31217584)
* Variant calls for samples in the PAGE study are part of TOPMED (dbGaP accession: phs001974.v5.p1; https://www.ncbi.nlm.nih.gov/projects/gap/cgi-bin/study.cgi?study_id=phs001974.v5.p1)
* GTEx v8 eQTL DAPG fine-mapping results (https://www.gtexportal.org/home/downloads/adult-gtex/qtl)

# Research involving human participants, their data, or biological material

Policy information about studies with human participants or human data. See also policy information about sex, gender (identity/presentation), and sexual orientation and race, ethnicity and racism.

| | |
|---|---|
| Reporting on sex and gender | Sex (as reported previously by the 1000 Genomes Project) was used as a covariate in all QTL mapping analyses. However, all samples are considered together, regardless of sex, for all analyses (i.e., analyses are not sex-stratified, but are sex-adjusted). |
| Reporting on race, ethnicity, or other socially relevant groupings | Samples were previously assigned to 1) populations and 2) continental groups (which comprise multiple populations) by the 1000 Genomes Project based on sampling location and expected patterns of genetic ancestry. We used these same continental group and population labels in this study.<br><br>For most analyses, samples were analyzed together, regardless of population label. For QTL mapping, the top 5 genotype PCs (which are found to correlate with population and continental group labels) were used as covariates to control for trans effects driven by global ancestry proportions. |
| Population characteristics | Initial sample collection was performed previously by the 1000 Genomes Project. All individuals were reported to be healthy adults at the time of sample collection. Blood was collected from each sample and was EBV-transformed to establish lymphoblastoid cell lines. 30x whole genome sequencing for each of these cell lines is available from the New York Genome Center, and we generated RNA-seq data from these cell lines in this study. |
| Recruitment | N/A - Recruitment performed as part of earlier study by the 1000 Genomes Consortium. |
| Ethics oversight | The Johns Hopkins Homewood IRB deemed this work not to meet the definition of human subjects research (HIRB00009187). |

Note that full information on the approval of the study protocol must also be provided in the manuscript.

# Field-specific reporting

Please select the one below that is the best fit for your research. If you are not sure, read the appropriate sections before making your selection.

☒ Life sciences ☐ Behavioural & social sciences ☐ Ecological, evolutionary & environmental sciences

For a reference copy of the document with all sections, see nature.com/documents/nr-reporting-summary-flat.pdf

# Life sciences study design

All studies must disclose on these points even when the disclosure is negative.

| | |
|---|---|
| Sample size | Sample size was chosen based on previous results by the GTEx consortium that demonstrated high power to detect eQTL associations at low MAF with 750 samples. We chose to sequence 780 libraries from 732 cell lines, such that we could sequence 24 cell lines in triplicate. The 732 cell lines were selected as evenly as possible from the 26 populations in the 1000 Genomes project. One library failed sequencing, leaving us with 779 total libraries across 731 unique cell lines. |
| Data exclusions | No data were excluded. |
| Replication | As described above, 24 cell lines were sequenced in triplicate. Library preparation and sequencing was successful for all replicates. Each of the replicated cell lines were sequenced twice in one sequencing batch, and a third time in a separate sequencing batch, to quantify within vs. between batch variation. We found that across the replicate libraries, the cell line that the library was generated from explained more variance in both expression level and splicing than the library batch (see Supplementary Information). |
| Randomization | Sequencing libraries were randomized across sequencing batches in a stratified manner: libraries were stratified across batches based on the population label of the cell line, to reduce confounding between sequencing batch and population. |
| Blinding | Not applicable: blinding was not relevant to experimental design or analysis. This is not a clinical study; there are no treatment or control groups. We are analyzing RNA-sequencing data from lymphoblastoid cell lines. |

# Reporting for specific materials, systems and methods

We require information from authors about some types of materials, experimental systems and methods used in many studies. Here, indicate whether each material, system or method listed is relevant to your study. If you are not sure if a list item applies to your research, read the appropriate section before selecting a response.

## Materials & experimental systems

| n/a | Involved in the study |
|---|---|
| ☒ | ☐ Antibodies |
| ☐ | ☒ Eukaryotic cell lines |
| ☒ | ☐ Palaeontology and archaeology |
| ☒ | ☐ Animals and other organisms |
| ☒ | ☐ Clinical data |
| ☒ | ☐ Dual use research of concern |
| ☒ | ☐ Plants |

## Methods

| n/a | Involved in the study |
|---|---|
| ☒ | ☐ ChIP-seq |
| ☒ | ☐ Flow cytometry |
| ☒ | ☐ MRI-based neuroimaging |

# Eukaryotic cell lines

Policy information about cell lines and Sex and Gender in Research

| | |
|---|---|
| Cell line source(s) | All cell lines are lymphoblastoid cell lines (LCLs) procured from the Coriell Insitute for Medical Research. Cell line sex is previously reported by the 1000 Genomes Project. |
| Authentication | Quality control procedures were performed according to standard practices of the NHGRI Repository at Coriell. Cell line identity was confirmed using a multiplex PCR assay for six autosomal microsatellite markers. |
| Mycoplasma contamination | At Coriell, cultures are tested and found free of mycoplasma, bacteria, and fungi during expansion, at the time of frozen storage, and after recovery of stock for distribution from liquid nitrogen. |
| Commonly misidentified lines (See ICLAC register) | Not applicable |

## Plants

Seed stocks

Not applicable

Novel plant genotypes

Not applicable

Authentication

Not applicable

