## [Peer Review File · Nature]

Manuscript Title: Sources of gene expression variation in a globally diverse human cohort

Reviewer Comments & Author Rebuttals

Reviewer Reports on the Initial Version:

Referees' comments:

Referee #1 (Remarks to the Author):

Taylor et al. generate and analyze bulk short-read RNA-Seq data and from 731 LCLs in 5 continental groups and 26 populations from 1000 Genomes together with previously generated high-coverage short-read whole genome sequencing data. This work has the potential to be an important resource for both human genetics/genomics and evolution researchers. Strengths of the study include their analysis showing frequency differentiated QTLs (fg-QTLs) are more frequent than heterogeneous effect QTLs (he-QTLs) when comparing populations, which could have implications for future PRS development and implementation. The authors perform standard eQTL/sQTL mapping, fine-mapping, and enrichment analyses. The paper is well written and the methodology is technically sound. Because there are limited functional genomics data in several included populations, these molecular QTL data will be of great use to human geneticists interested in pinpointing causal variants underlying disease and understanding genome function across populations. While a valuable resource, it is somewhat limited by the one cell line type and limited African diversity. I have several questions about the methodological approach and suggestions for improvement:

1. In your methods, you describe using 60 PEER factors for eQTL mapping and 15 PEER factors for sQTL mapping. Why the difference? Did you evaluate whether these PFs associate with known confounders or assess a range of PFs to maximize QTL discovery? How might your populations/tissue differ from GTEx in this regard?
2. The ages of the LCLs are known to vary, i.e., the CEU lines are decades older than some of the 1000 Genomes lines, e.g., <https://www.nature.com/articles/srep07960>. Did you see any evidence of confounding by age of cell lines in your differential expression analyses?
3. The ADMIXTURE plot in Fig 1B is misleading since most of the "African" samples across all studies presented are "West African" and makes it look like there is less diversity in Africa than other continents. While I don't think you can or should cater your presentation to how racists may interpret your results, see some examples in <https://www.nature.com/articles/d41586-022-03252-z>, the negative effects of presenting ancestry in such a discrete manner is something to consider. An alternative could be a figure like Fig S2 with proportion variance explained by each PC included, which would better reflect the continuum of genetic diversity.
4. Since MAGE is meant to be a resource, you should include X chromosome QTLs in your analyses.

5. Adding a colocalization analysis of your fine-mapped QTLs with common disease/biomedical trait GWAS would be helpful to quantify the benefits of MAGE to pinpointing causal effects, especially in comparison to EUR-only QTLs.

6. I agree with the authors that a strength of LCLs is that environmental variation is reduced compared to other tissues, which could be one reason he-QTLs are rare in LCLs, which the authors discuss in the context of future dynamic QTL studies, starting at line 327. I also wonder how generalizable your results are to other complex traits, given that the local/cis genetic architecture of gene expression is sparse (usually 1-2 credible sets), whereas most common diseases are much more polygenic (more, but smaller effect sizes). Can you comment on the feasibility of focusing on causal signals that do not make assumptions about the number of causal variants at a locus for much more polygenic traits? What are the implications for PRS implementation and population portability in the context of your study and Mostafavi et al. (ref. 42)?

7. The authors did a great job organizing their github repo, which makes summary results publicly available. Consider adding the summary data to zenodo, with its persistent doi, rather than just keeping in Dropbox.

Minor

1. A Figure like Fig 5A showing variant counts rather than cumulative fraction could be useful to emphasize greater genetic diversity in the African populations.

2. Delete the "a" in "a untyped causal variants", line 293.

3. I'd delete "previously" in the abstract's "private to previously underrepresented populations." These populations are still underrepresented.

I commend the authors for their comprehensive effort in undertaking this project in understudied populations and pushing human genetics forward. Thank you for presenting your results in a well-organized fashion.

Referee #2 (Remarks to the Author):

In the manuscript entitled “Sources of gene expression variation in a globally diverse human cohort”, Taylor et al. perform an expression and splicing QTL study on LCLs of 731 individuals of diverse ancestries whose genomes have been sequenced by the 1KG study. They discover a large number of eQTLs and sQTLs. They perform fine-mapping with Susie allowing for multiple credible sets per gene. They find that a small proportion of eQTLs have a significant interaction with ancestry, and of these most involve eGenes with more than one credible set. These ancestry-specific effects disappear when controlling for the other independent eQTLs in the gene. Together these results suggest that effect sizes for cis eQTLs vary little among ancestral groups in LCLs.

The manuscript is clear and well written. In addition, the authors did a fantastic job in making this resource accessible, with many tables with their intermediate results available in dropbox and the raw RNA-seq data available in SRA. I believe this will be a useful resource for the human genetics community.

Main concerns:

- Even though controlling for the top 5 genetic PCs makes sense for identifying eQTLs in general, I wonder whether this could be biasing the analysis of SNP by ancestry interactions? If I were looking for an eQTL interaction with a certain variable, I wouldn't control for that variable when first identifying a set of eQTLs in which to test the interaction. Work from Andrew Clark's lab, Marderstein et al. AJHG 2021, shows that if you want to find GxE eQTL interactions, you can first identify variance QTLs (which are enriched for GxE interactions) and then test the interaction of interest in those vQTLs. I wonder if the authors can adopt a similar strategy to try to look more purposely for those possibly existing ancestry-eQTL interactions?
- Also, I believe that in the section 17 of the Methods, Model 1 should have the Xcg variable included, and Model 2 should have the Xcg variable alone too, apart from the Xcg*Gji that is already included. That way when you compare the two models you are specifically testing for the interaction term.
- The authors did a great job at characterizing eQTLs, but splicing QTLs remain not that well characterized in the manuscript, including enrichment analysis for functional/regulatory annotations, and the ancestry interaction analyses.
- The authors depict in Figure 1 how diverse their cohort is compared to other eQTL studies such as GTEx, Geuvadis and AFGR. However, the authors failed to cite the study by Stranger et al. Plos Genetics 2012, in which they had a similar sample size and used LCLs from eight global populations of the HapMap3 project that included populations not present in GTEx, Geuvadis and AFGR. This new MAGE study uses RNA-seq and WGS instead of microarray expression and microarray genotypes, and more sophisticated analyses are made with new tools available, but the main results remain very similar to the Stranger et al study.

Minor: In Figure 5A, there is no legend for: C, R, U.

Referee #3 (Remarks to the Author):

In this manuscript, Taylor et al. present a new resource comprising RNA-seq data from 731 globally distributed individuals of the Thousand Genomes Project. They additionally evaluate the geographic distribution of variation in gene expression and splicing, map signals to putatively causal loci, assess the magnitude of effect sizes across populations for causal variants, and comment on the evolutionary implications of their findings. I believe this resource will be highly valuable to the field, complementing GTEx in its current use cases, and ultimately will allow for improved ancestral generalizability of future work evaluating gene expression. The researchers indeed highlight here the value of including more diverse representation in expression and splicing analyses, finding a healthy number of both eQTLs and sQTLs that are restricted to an understudied population. I believe the analyses presented here are sufficient to tell the story, but have a few questions for the authors regarding potential analytic concerns, and think that the text could use some expansion to aid the reader in appreciating the value of this work.

Comments:

1. How concerned should we be about potential reference bias? That is, mapping to existing traditional reference genomes will work better for populations that are better represented in the construction of those genome builds. How big of an impact would this be on your results, as it could lead to poorer evaluation of expression in the underrepresented populations included here?
2. The introduction is quite brief. More background on prior studies on expression across populations, eQTLs, sQTLs would be helpful to set the scene for this work and clarify the current state of the field.
3. Similarly, the discussion is again pretty succinct. In particular, expanding on the utility of the novel loci discovered here in understudied populations, how inclusion of more representative samples will improve ancestral generalizability of findings, the potential downstream impacts of this resource on improving personalized medicine, etc. would highlight the value of this resource.
4. It was unclear to me how the presence of duplicated samples presented in the ADMIXTURE plot was handled. The methods seem to imply that one mega ADMIXTURE run of all samples was conducted, which would slightly bias ancestry estimation if some samples are included multiple times (e.g. the same YRI are showing up in MAGE, Geruvadis, and AFGR?). This shouldn't qualitatively change the results drastically, but would affect the global proportion estimates.
5. The authors point out the lack of population labels in GTEx. Given their analyses here, they are able to infer ancestry for each participant, though. It would be very useful if the authors could provide population assignments for GTEx samples based on their work for use in future studies. I cannot speak to any administrative challenges in terms of interfacing with GTEx, but is this logistically possible?
6. The authors cite their findings as indicative of negative and stabilizing selection at different points in the manuscript. This is not wrong per se, but a bit of explanation for readers not embedded in evolutionary genomics might help unpack the interpretation of how both modes of selection can be invoked. E.g., evidence for maintenance of expression levels = stabilizing selection, but the constraint on DNA sequence underlying the eQTL used to support this argument is actually from negative selection removing sequence variation.

7. Also regarding evolution, could the authors utilize more localized selection statistics to assess the impact of negative selection on expression related loci? Currently they are just using the gene-level metric pLI, but this would miss any nuance of the particular area of a gene that might be under strongest evolutionary constraint. Zooming in to narrower areas around the causal variant to show signals of elevated negative selection would be even more convincing than a gene-wide score.

Minor comments:

1. Figure 1 legend – ADMIXTURE plots are described as presenting ‘hypothetical’ ancestry components. I think a better word choice could be used – inferred would be more technically correct.
2. The dashed lines in figure 3 are not the most intuitive (the line looks like a threshold definition on first glance), though I appreciate what the authors are trying to accomplish by including them. A different way to visualize this might be better... perhaps a bracket?
3. There are some font/formatting issues to be fixed.

Author Rebuttals to Initial Comments:

Response to Referee Comments

We thank the editor and referees for their positive reception of our study, as well as their constructive comments and suggestions that we believe have substantially improved our manuscript. These critiques helped us bolster and better contextualize our findings, while also improving the utility of MAGE as a resource for studying human gene expression diversity. We highlight several major additions:

- An expanded introduction summarizing several previous studies that are directly relevant to and provide context for this work
- The addition of chrX *cis*-e/sQTLs to the published data resource
- A more thorough investigation of the functional impacts of sQTLs
- A colocalization analysis characterizing shared signals between biomedical/disease trait GWAS in non-European cohorts and MAGE *cis*-e/sQTLs
- An auxiliary analysis of eQTL effect size heterogeneity between continental groups (i.e., he-eQTLs) that addresses some of the potential biases of the primary he-eQTL analysis
- An expanded discussion section that considers the implications of our finding that molecular trait effect sizes are consistent across groups for more polygenic complex traits and highlights the added benefit of a diverse gene expression dataset like MAGE in better understanding the molecular mechanisms of human traits, particularly in historically underrepresented populations, as well as opportunities for future research

The comments from the referees are copied below in black text and are organized by major and minor comments from each referee. Our responses are indicated with interspersed blue text. Where appropriate, we also copied text/figures from the manuscript that were added/changed in response to the referee's comments. We have also submitted a version of the manuscript with "track changes" that annotates all revisions.

Referee #1: Genetic Regulation

Taylor et al. generate and analyze bulk short-read RNA-Seq data and from 731 LCLs in 5 continental groups and 26 populations from 1000 Genomes together with previously generated high-coverage short-read whole genome sequencing data. This work has the potential to be an important resource for both human genetics/genomics and evolution researchers. Strengths of the study include their analysis showing frequency differentiated QTLs (fg-QTLs) are more frequent than heterogeneous effect QTLs (he-QTLs) when comparing populations, which could have implications for future PRS development and implementation. The authors perform standard eQTL/sQTL mapping, fine-mapping, and enrichment analyses. The paper is well written and the methodology is technically sound. Because there are limited functional genomics data in several included populations, these molecular QTL data will be of great use to human geneticists interested in pinpointing causal variants underlying disease and understanding genome function

across populations. While a valuable resource, it is somewhat limited by the one cell line type and limited African diversity. I have several questions about the methodological approach and suggestions for improvement:

Thank you for your positive response to our manuscript and for the pertinent questions and valuable suggestions for improving the utility of the MAGE resource.

Major Comments:

Major Comment #1

In your methods, you describe using 60 PEER factors for eQTL mapping and 15 PEER factors for sQTL mapping. Why the difference? Did you evaluate whether these PFs associate with known confounders or assess a range of PFs to maximize QTL discovery? How might your populations/tissue differ from GTEx in this regard?

These are all very good questions. The number of PEER factors chosen for e/sQTL mapping were based on optimizations previously performed by the GTEx consortium (GTEx Consortium, Nature. 2017; GTEx Consortium, Science. 2020). Briefly, for each of four sample size bins, GTEx assessed the number of PEER factors that maximized the number of significant e/sGenes discovered. As a result of these optimizations, GTEx used 60 PEER factors for eQTL mapping with sample sizes ≥ 350 and 15 PEER factors for sQTL mapping regardless of sample size. Based on the optimizations by GTEx and the sample size of MAGE, we chose to use 60 PEER factors for eQTL mapping and 15 PEER factors for sQTL mapping.

We recognize that the PEER factors identified in GTEx may differ from those in MAGE. As the reviewer noted, 1) GTEx tissues may include multiple cell types while MAGE comprises only a single cell type, and 2) GTEx largely comprises individuals of European ancestries, while MAGE spans a broad range of human populations. Additionally, the RNA library preparation and sequencing procedures are not identical between the two studies.

To address the reviewer's comment, we therefore investigated how the number of e/sGenes varies according to the number of PEER factors within MAGE, restricting analysis to Chromosome 1 for computational efficiency. For eQTL mapping, we discovered eGenes using the FastQTL adaptive permutation mode (the same procedure used in the main text) with either 0, 1, 2, 5, 10, 15, 20, 25, 30, 40, 50, 60, 70, 80, or 100 PEER factors. Similarly, for sQTL mapping optimization, we discovered sGenes with either 0, 1, 2, 5, 10, 15, 20, 25, 30, 40, or 50 PEER factors. The results are shown below:

Figure S9. Selection of PEER factors for QTL mapping. Number of eGenes and sGenes on Chromosome 1 discovered with FastQTL using different numbers of PEER factors as covariates. e/sGenes were discovered at a 5% FDR. For eQTL mapping, PEER factors were computed from normalized TMM values from the autosomes and chrX (the same values used as input for eQTL mapping). For sQTL mapping, PEER factors were computed from normalized intron excision ratios from the autosomes and chrX (the same values used as input for sQTL mapping).

For eQTL mapping, we observe a plateau starting at roughly 50 PEER factors whereafter adding PEER factors does not substantially increase the number of eGenes discovered. For sQTL mapping, the number of sGenes discovered is relatively robust to the number of PEER factors used as covariates. We emphasize that there is no “ground truth” for optimization, and choosing the number of PEER factors to maximize e/sGene discovery within our data may risk overfitting. That said, our choices of 60 and 15 PEER factors respectively for e/sQTL mapping are consistent with GTEx and fall near the plateaus/peaks of e/sGene discovery, and we therefore elected to maintain these selections for our analyses.

We have integrated this figure in the Supplementary Material (**Fig. S9**) and have clarified our reasoning for choosing these numbers of PEER factors in Supplemental Methods section **10.3**.

We also observed that the selected PEER factors are correlated with known technical covariates from the RNA-seq study. Specifically, we tested for correlation between the PEER factors and eight technical variables: sequencing batch, the study in which the cell lines were first generated (i.e., the International HapMap Project, HapMap 3, or the 1000 Genomes Project), continental group, population, sex, RNA integrity number (RIN), the total number of reads, and the total amount of RNA in the library. We also requested a metric of cell line age from Coriell, where the cell lines were purchased, but Coriell declined to provide these dates. Nevertheless, we expect these ages to correlate with the study in which the cell lines were generated. Our complete set of correlation results is shown in the figure below:

Figure S10. Correlation between PEER factors and known confounders. For each of the 60 expression-derived PEER factors and 15 splicing-derived PEER factors, we tested their correlation with each of eight possible confounders. **(A)** Correlation between the 60 expression-derived PEER factors and the tested confounders. **(B)** The significance of each of the correlations shown in A. Non-significant correlations (based on a Bonferroni threshold of 8.33×10^{-5}) are shown in grey. **(C)** As in A, but for the 15 splicing-derived PEER factors. **(D)** As in B, but for the 15 splicing-derived PEER factors.

For expression-derived PEER factors we observed the strongest (**Panel A**) and most significant (**Panel B**) correlations with sequencing batch. The same pattern is observed in the splicing-derived PEER factors (**Panels C and D**). We also note significant but weaker correlations with continental group and population for both expression- and splicing-derived PEER factors, supporting their exclusion in the analyses presented in **Figure 2**.

We have integrated this figure in the Supplementary Material (**Fig. S10**) and have briefly summarized these patterns in Supplemental Methods section **10.3**.

Major Comment #2

The ages of the LCLs are known to vary, i.e., the CEU lines are decades older than some of the 1000 Genomes lines, e.g., <https://www.nature.com/articles/srep07960>. Did you see any evidence of confounding by age of cell lines in your differential expression analyses?

This is a pertinent question, and we are aware that factors such as cell line age can be confounders in RNA-seq datasets. While we were unable to obtain ages of the cell lines from Coriell, we did test for correlations between several other potential confounders (including populations and study of origin, both of which we expect to correlate with cell line age) and the PEER factors used as covariates in our analyses. The results of this analysis are discussed in detail in response to the previous question, but briefly we do observe weak correlations between our PEER factors and population and study of origin. That said, the strongest correlations were with sequencing batch, which we explicitly control for in our analyses partitioning expression variation within and between populations.

Major Comment #3

The ADMIXTURE plot in Fig 1B is misleading since most of the “African” samples across all studies presented are “West African” and makes it look like there is less diversity in Africa than other continents. While I don’t think you can or should cater your presentation to how racists may interpret your results, see some examples in <https://www.nature.com/articles/d41586-022-03252-z>, the negative effects of presenting ancestry in such a discrete manner is something to consider. An alternative could be a figure like Fig S2 with proportion variance explained by each PC included, which would better reflect the continuum of genetic diversity.

This is an important point, and we also debated the use of ADMIXTURE versus PCA and/or other visualizations, given their potential to over-emphasize genetic differences and obscure the high levels of variation within Africa. Despite these limitations, the ADMIXTURE plot demonstrates the inclusion of South Asian, Admixed American, and East Asian ancestry groups in a way that is difficult to glean from the PCA plot. To address this comment, we have therefore integrated the PCA and ADMIXTURE plots as separate panels in Figure 1, while also displaying the proportion of variance explained by each PC, as suggested. We additionally expanded the figure legend to state:

"Ancestry components are modeling constructs that do not directly correspond to true ancestral populations, and the results of ADMIXTURE analysis strongly depend on sampling characteristics of the input data. While $k=7$ minimizes the cross-validation error within this combined data set (Fig. S1), alternative choices of k reflect structure at different scales (Fig. S2)."

The cited supplemental figures include a new set of ADMIXTURE plots (Fig. S2) where we vary k from 3 to 9, which highlights the dependence of these visual summaries on the selection of k . We also added a sentence to the main text (where this figure is referenced) to emphasize the greater genetic diversity in Africa and cite the related AFGR study that focuses on African diversity:

"While we emphasize the greater genetic diversity within African populations—a point largely obscured by ADMIXTURE and principal component analyses²⁰—these visualizations demonstrate that our study includes data from several non-African ancestry groups that were poorly represented in previous studies (Fig. 1B, 1C, 1D, Fig. S2, S3; also see the African Functional Genomics Resource [AFGR]¹⁶ for a related study of African populations)."

Finally, we expanded the section of the Discussion noting the limited sampling of African populations:

"Finally, while geographically diverse, the sampling of IKGP is not without biases—for example, narrowly sampling the vast diversity within Africa and excluding indigenous

populations from Oceania and the Americas, as well as countless other populations. Addressing these biases will require deeper community engagement and respect for the rights, interests, and expectations of research participants from diverse human groups⁵⁶. This expansion of diversity in functional genomics parallels efforts for improved representation of diversity in genome sequencing and assembly, including construction of pangenomes^{57,58}. While the current study was based on alignment to a linear representation of the reference genome, given the maturity of software tools and annotations built on this paradigm, MAGE offers an ideal data resource for testing pangenomic methods over the coming decade as they are developed by the research community."

Major Comment #4

Since MAGE is meant to be a resource, you should include X chromosome QTLs in your analyses.

Thank you for this suggestion. We agree that inclusion of QTLs from the X chromosome would improve the utility of the MAGE resource. We do note, however, that there are some caveats to performing QTL mapping on the X chromosome and interpreting results. First, we assume the pseudoautosomal regions (PARs) on the X chromosome represent both the X chromosome PARs and Y chromosome PARs. This allows us to discover QTLs in the X-PARs exactly the same way we did so on the autosomes. In the X chromosome non-PARs, however, XX individuals are diploid, while XY individuals are haploid. To enable joint eQTL mapping with XX and XY samples in the non-PARs, we artificially transform XY haploid genotypes to homozygous diploid genotypes in the input VCF file.

Altogether, this approach for handling the X chromosome makes two important assumptions about the architecture of gene expression on the X chromosome: 1) genes in the PARs “escape” X-inactivation (i.e. both the X and Y homologs of each gene are expected to be expressed) and 2) genes in the non-PARs are randomly inactivated on one X chromosome homolog in XX samples and are not inactivated in XY samples. While the first assumption is expected to be valid for most genes, some non-PAR genes are known to escape X-inactivation.

Using this approach, we performed e/sQTL mapping and fine-mapping for genes on the X chromosome. We also estimated effect sizes for eQTLs in the X-PARs, but were unable to do so for eQTLs in the X chromosome non-PARs because the aFC-n tool does not natively handle haploid genotypes and there is not a straight-forward method to transform genotypes in a way that provides meaningful effect size estimates for these eQTLs.

We have detailed the approach described above in the supplemental methods (largely section **10.4**). However, given the caveats of interpreting QTL results on the X chromosome, we continue to focus our QTL analyses in the main text on the autosomes. That said, we include the

full set of X chromosome e/sQTL results in our published data along with the results from the autosomes.

Major Comment #5

Adding a colocalization analysis of your fine-mapped QTLs with common disease/biomedical trait GWAS would be helpful to quantify the benefits of MAGE to pinpointing causal effects, especially in comparison to EUR-only QTLs.

This is a terrific suggestion and an analysis we were excited to incorporate into the paper. While GWAS of non-European populations are limited, one cohort that we thought would facilitate an assessment of the utility of MAGE in interpreting complex trait GWAS in non-European populations is that of the Population Architecture using Genomics and Epidemiology (PAGE) study. PAGE comprises 49,839 non-European individuals, including large samples of individuals who self-reported as Hispanic/Latin American or African American, as well as smaller samples of individuals who self-reported as Asian, Native Hawaiian, or Native American. As such, MAGE provides a unique resource for interpreting the molecular underpinnings of the complex traits measured in PAGE.

We performed colocalization analysis to identify shared signals between GWAS of 25 complex traits from PAGE (including quantitative biomedical traits and disease phenotypes) and *cis*-e/sQTLs from MAGE. We discovered moderate-to-strong evidence of colocalization with MAGE eQTLs for 39 independent GWAS signals across 14 traits in PAGE, and moderate-to-strong evidence of colocalization with MAGE sQTLs for 30 independent GWAS signals across 12 traits in PAGE. Critically, these include GWAS signals that are rare ($MAF < 0.05$) or unobserved in European groups, demonstrating the unique added benefit of using a diverse resource like MAGE for interpreting GWAS in non-European populations.

These results have been incorporated into the relevant main text of the manuscript in more detail. The full analysis is described in the Supplementary Methods (section 14), and the results are summarized in **Fig. S21**.

Major Comment #6

I agree with the authors that a strength of LCLs is that environmental variation is reduced compared to other tissues, which could be one reason he-QTLs are rare in LCLs, which the authors discuss in the context of future dynamic QTL studies, starting at line 327. I also wonder how generalizable your results are to other complex traits, given that the local/*cis* genetic architecture of gene expression is sparse (usually 1-2 credible sets), whereas most common diseases are much more polygenic (more, but smaller effect sizes). Can you comment on the feasibility of focusing on causal signals that do not make assumptions about the number of causal variants at a locus for much more polygenic traits? What are the implications for PRS

implementation and population portability in the context of your study and Mostafavi et al. (ref. 42)?

These are terrific questions and we agree that these topics warrant more discussion in the manuscript. Regarding the genetic architecture of gene expression versus common diseases, we suspect that the seemingly small number of causal signals for gene expression is driven in part by lower power in our analysis compared to previous complex trait GWAS. As sample size in QTL studies increases to the sample sizes of complex trait GWAS, we would expect to reveal even more allelic heterogeneity. Additionally, eQTL mapping studies in African populations (e.g. DeGorter et al. 2023, doi: 10.1101/2023.11.04.564839)—which are known to have more genetic diversity—have found as many as 6 credible sets per gene on average.

That said, we recognize that complex traits can be especially polygenic, and the question of whether our approach is suitable for these more polygenic traits is an interesting one. A recent paper by Hou et al. (doi: 10.1038/s41588-023-01338-6) investigated effect size heterogeneity in admixed individuals across 38 complex traits in three different studies. The authors concluded that GWAS nominal effect sizes are consistent across local ancestries, and that apparent deviations can be explained by multiple causal variants within a region, consistent with our own conclusions in this manuscript in regards to the genetic architecture of gene expression. This result is promising and suggests that our strategy of focusing on causal signals without assumptions about the number of such signals at a locus may work well with GWAS. We are excited to see how future GWAS in diverse cohorts apply this methodology and assess its impact on portability. We have added text to this effect to the discussion section of the manuscript.

"The scale and diversity of the data set enabled the discovery of numerous potentially novel genetic associations, while offering high resolution for identifying putatively causal variants and elucidating their mechanisms of action. Our study also demonstrates that conditional on the correct identification of causal variants, the effects of such variants tend to be additive and highly consistent across populations, addressing a point of recent debate within the field^{7,14,39,40,47}. This observation in turn suggests that for gene expression, ancestry-dependent epistatic effects tend to be weak and/or rare in human genomes, in contrast to some observations from other model systems⁴⁸. The extent to which our approach of focusing on causal signals while allowing for multiple causal signals at a locus is suitable for more polygenic complex traits is an open question, but a recent study that investigated effect size heterogeneity in admixed individuals across 38 complex traits found that nominal effect sizes are consistent across local ancestries³⁹. Apparent deviations could be explained by multiple causal variants within a region, consistent with our own conclusions. This result is promising and suggests that our strategy may work well with GWAS of more polygenic complex traits. Such consistency of genetic effects further motivates the use of diverse samples for association studies, as a common causal variant identified in one population may inform the effect of that variant in a population where the same variant is rare and association testing would be underpowered. Because effects tend to be consistent across populations, the functional effect should be similar in the population where the

allele is rare. This has direct implications for downstream predictive applications like PRS and for personalized medicine, as inclusion of these population-stratified causal variants in predictive models should lead to more accurate and generalizable predictions⁴⁹. Thus all populations—not only underrepresented populations—benefit from the inclusion of greater diversity in genetic studies."

In regard to Mostafavi et al., this approach may not significantly improve colocalization between complex trait GWAS and molecular QTL studies. The authors of that study found that limited colocalization stems from distinct selective pressures shaping genetic variation that can be identified in the two analyses: GWAS hits tend to occur within genes under strong purifying selection, whereas molecular QTLs are most easily identified for genes under relaxed constraint. We have added text about this phenomenon to the discussion section of the manuscript.

"Intersection of e- and sQTLs with data from GWAS may facilitate understanding of the molecular mechanisms linking genetic variation to organismal phenotypes. Using GWAS data from the PAGE study of ancestrally diverse individuals, we identified 54 GWAS signals that colocalize with e- and/or sQTL signals. While informative and substantial in absolute number, these reflect a minority of all GWAS hits. Limited colocalization between molecular QTLs and GWAS hits is well-described, largely stemming from distinct selective pressures shaping genetic variation that can be identified (with incomplete statistical power) in the two analyses⁵⁰. GWAS hits tend to occur within genes under strong purifying selection, whereas molecular QTLs are most easily identified for genes under relaxed constraint. This is consistent with our finding that genes exhibiting strong signals of selection are depleted of MAGE eQTLs. While inclusion of additional tissues and cell types modestly increases the rate of colocalization, these qualitative observations hold even for multi-tissue studies such as GTEx⁵⁰. Despite these general limitations of colocalization analysis, our results demonstrate cases where MAGE facilitates interpretation of GWAS results, particularly in underrepresented populations. We anticipate that this utility will further improve as GWAS continue to expand to more diverse cohorts."

Major Comment #7

The authors did a great job organizing their github repo, which makes summary results publicly available. Consider adding the summary data to zenodo, with its persistent doi, rather than just keeping in Dropbox.

We agree with this important point and have deposited the processed data (e/sQTL calls, fine-mapping results, expression and splicing matrices, etc.) on Zenodo as suggested (doi: 10.5281/zenodo.10535719). Raw data is available on SRA (accession: PRJNA851328).

Minor Comments:

Minor Comment #1

A Figure like Fig 5A showing variant counts rather than cumulative fraction could be useful to emphasize greater genetic diversity in the African populations.

Thank you for this suggestion. It should be noted that for this analysis we did not perform eQTL mapping in each continental group separately, but across the entire MAGE dataset at once (adjusting for global ancestry using top genotype principal components as covariates). Figure 5A is therefore meant to highlight how a larger *proportion* of the eQTLs in MAGE are geographically restricted relative to GTEx, especially from underrepresented populations (**Fig. S22** [previously **S17**] also supports this claim).

The reviewer's comment about greater genetic diversity in Africa is nevertheless valid, and we have therefore addressed it explicitly in relation to Figure 1 where it is also relevant: *“While we emphasize the greater genetic diversity within African populations—a point largely obscured by ADMIXTURE and principal component analyses²⁰—these visualizations demonstrate that our study includes data from several non-African ancestry groups that were poorly represented in previous studies (Fig. 1B, 1C, 1D, Fig. S2, S3; also see the African Functional Genomics Resource [AFGR]¹⁶ for a related study of African populations).”*

Minor Comment #2

Delete the “a” in “a untyped causal variants”, line 293

Thank you! This change has been made.

Minor Comment #3

I'd delete “previously” in the abstract's “private to previously underrepresented populations.” These populations are still underrepresented.

We agree, and this change has been made.

I commend the authors for their comprehensive effort in undertaking this project in understudied populations and pushing human genetics forward. Thank you for presenting your results in a well-organized fashion.

Thank you very much for these encouraging words, as well as the thorough and constructive critiques that greatly improved our work.

Referee #2: Human genetics, QTL mapping

In the manuscript entitled “Sources of gene expression variation in a globally diverse human cohort”, Taylor et al. perform an expression and splicing QTL study on LCLs of 731 individuals of diverse ancestries whose genomes have been sequenced by the 1KG study. They discover a large number of eQTLs and sQTLs. They perform fine-mapping with Susie allowing for multiple credible sets per gene. They find that a small proportion of eQTLs have a significant interaction with ancestry, and of these most involve eGenes with more than one credible set. These ancestry-specific effects disappear when controlling for the other independent eQTLs in the gene. Together these results suggest that effect sizes for cis eQTLs vary little among ancestral groups in LCLs.

The manuscript is clear and well written. In addition, the authors did a fantastic job in making this resource accessible, with many tables with their intermediate results available in dropbox and the raw RNA-seq data available in SRA. I believe this will be a useful resource for the human genetics community.

We thank the reviewer for this positive response to our study, as well as the insightful suggestions that follow.

Major Comments:

Major Comment #1

Even though controlling for the top 5 genetic PCs makes sense for identifying eQTLs in general, I wonder whether this could be biasing the analysis of SNP by ancestry interactions? If I were looking for an eQTL interaction with a certain variable, I wouldn't control for that variable when first identifying a set of eQTLs in which to test the interaction. Work from Andrew Clark's lab, Marderstein et al. AJHG 2021, shows that if you want to find GxE eQTL interactions, you can first identify variance QTLs (which are enriched for GxE interactions) and then test the interaction of interest in those vQTLs. I wonder if the authors can adopt a similar strategy to try to look more purposely for those possibly existing ancestry-eQTL interactions?

We thank the reviewer for their feedback regarding our he-eQTL discovery analysis. We want to clarify that our interaction analysis is testing for genotype-by-continental-group interactions, rather than genotype-by-ancestry interactions, although we acknowledge that continental group labels and various definitions of genetic ancestry may be correlated. Regarding the reviewer's comment about bias being introduced in the interaction test by including top 5 genetic PCs as covariates in the initial eQTL mapping step, we note that the top 5 genetic PCs are only included as main effect variables in the regression (effectively allowing for differential expression among ancestry groups, independent of genotype) and do not preclude the existence of any interaction effects (i.e., differences in the slope among continental groups).

The reviewer’s suggestion to instead first identify vQTLs and then test for genotype-by-continental-group interactions in this discovery set (à la Marderstein et al. 2021) is an intriguing one, but we were concerned that this approach may lead to false positive interactions caused by failure to account for the additive effects of multiple causal variants. To illustrate this point, we simulated gene expression for a gene with two causal variants: “Var1” and “Var2”, shown in the figure below. Our simulation comprises two separate populations: “Pop1” and “Pop2”. Var1 is common in both populations, while Var2 is common in Pop1 but is unobserved in Pop2 (**Panel A** below). Additionally, within Pop1, Var1 and Var2 are in strong LD ($R^2 = 0.81$). Critically, while Var1 and Var2 have different effect sizes, *these effects are consistent across populations* (**Panel B**). We simulate expression Y_i of individual i as $Y_i \sim N(X_{1i}\beta_1 + X_{2i}\beta_2, I)$, where X_{1i} and X_{2i} describe the genotypes (0, 1, or 2) of Var1 and Var2, respectively, for individual i . β_1 and β_2 describe the effect sizes of Var1 and Var2, respectively.

Focusing solely on Var1, we observe a pattern consistent with a vQTL and that Var1 exhibits a stronger measured effect size in Pop1 than Pop2 (**Panel C**). Supporting this hypothesis, the deviation regression proposed by Marderstein et al. yields a highly significant result (**Panel D**). However, we know that the simulation does not include a true SNP-by-population interaction. Indeed, regressing out the effect of Var2 removes the apparent effect size heterogeneity (**panels E,F**).

That said, we agree that our current eQTL discovery pipeline may be biased toward identifying eQTLs whose effect size does not differ between continental groups. This bias could arise because one of the inherent assumptions of standard eQTL mapping and fine-mapping is that each causal variant has a single effect size that does not vary between subsets of the sample. As we only test fine-mapped variants for effect size heterogeneity, this assumption may diminish our sensitivity for identifying he-eQTLs.

To address these concerns, we performed eQTL mapping, fine-mapping, and effect size estimation separately within each of the continental groups represented in MAGE. For the resulting credible sets, we then compared effect sizes between continental groups to ask if the

effects of causal variants remain consistent, even when estimated independently in each continent.

For each pair of continental groups, we considered two credible sets comparable if they shared at least one variant and corresponded to the same gene (detailed in the Supplemental Methods). After Bonferroni correction, we found that between 97.5% (comparisons between AFR-EUR) and 99.8% (comparisons between AMR-SAS) of credible sets did not have significantly different effect sizes (Welch's t-test). These results corroborate our original finding that effect size heterogeneity is rare among eQTLs and show that this is true even when explicitly allowing effect sizes to vary between continental groups during eQTL discovery.

Figure S28. High concordance in credible set effect sizes across continental groups. (A) Heatmap showing the fraction of shared causal signals where $\log_2(aFC)$ is not significantly different in pairs of continental groups after Bonferroni correction. *n* represents the total number of shared causal signals in each pair of continental groups. (B) Scatterplots comparing $\log_2(aFC)$ within pairs of continental groups. Points are colored by whether the effect sizes are significantly different. The black line plots $y = x$ (i.e. theoretical identical effect sizes). The gray line plots the best fit linear trendline.

We have updated the Supplemental Methods with a description of this continental group-stratified QTL mapping analysis, as well as the Results section of the paper, and this figure has been added to the supplement (Fig. S28): “An alternative approach based on stratified eQTL mapping and fine-mapping within each continental group (see Methods) likewise indicated high consistency in effect sizes, such that 97.5-99.8% of credible sets had similar effect sizes between pairs of continental groups (Fig. S28).”

Major Comment #2

Also, I believe that in the section 17 of the Methods, Model 1 should have the Xcg variable included, and Model 2 should have the Xcg variable alone too, apart from the Xcg*Gji that is

already included. That way when you compare the two models you are specifically testing for the interaction term.

Thank you for catching this! We note that the model was run correctly (i.e., as the reviewer specified), and section 17 (now section 18) of the supplemental methods simply had a typo. This typo has been corrected.

Major Comment #3

The authors did a great job at characterizing eQTLs, but splicing QTLs remain not that well characterized in the manuscript, including enrichment analysis for functional/regulatory annotations, and the ancestry interaction analyses.

We thank the reviewer for this suggestion; we agree that a more thorough characterization of the sQTLs we discovered is warranted and will improve the utility of the MAGE resource. To investigate the functional implication of our discovered sQTLs, we annotated fine-mapped lead sQTLs with the ensembl Variant Effect Predictor (VEP) tool with a total of 26 genomic annotations. We also annotated a randomly selected set of “null” variants (matched for TSS distance and MAF with the lead sQTLs) to serve as a comparison. We observed a strong enrichment of lead sQTLs in key splicing-relevant annotations relative to the matched null. These findings highlight the biological relevance of our fine mapped cis-sQTLs in splicing regulation and their likely functional impact on the splicing processes. This analysis has been described in detail in the Supplementary Methods (section 13.2) and the results are now integrated into the relevant main text of the manuscript and **Fig. 4D**, which are also copied below:

*“We also investigated the genomic context of our fine-mapped cis-sQTLs. We observed strong enrichment of lead sQTLs in several key splicing-relevant annotations including splice donor sites ($\log_2(\text{fold enrichment}) = 6.07$, 95% CI [4.09, 8.04]) splice acceptor sites ($\log_2(\text{fold enrichment}) = 5.52$, 95% CI [3.54, 7.50]), and nearby regions ($\log_2(\text{fold enrichment}) = 4.15$, 95% CI [3.70, 4.62]) at intron-exon boundaries (**Fig. 4D**). Despite their magnitude of enrichment, variants in canonical splice sites and splice regions represented a minority of lead sQTLs, with a greater abundance of sQTLs falling within 5' and 3' UTRs, as well as exons of both coding and non-coding genes. While exhibiting weaker enrichments, these annotation categories together cover a much larger mutational target size and may encompass splicing enhancers and cryptic splice sites for which annotation remains an open challenge. In contrast, intergenic regions were strongly depleted of lead sQTLs, despite matching our null background variant sets on minor allele frequency (MAF) and distance from the TSS ($\log_2(\text{fold enrichment}) = -2.51$, 95% CI [-2.58, -2.43]; see Methods). These findings support the biological validity of the fine-mapped cis-sQTLs and lend insight into the mechanisms by which these variants impact splicing.”*

Figure 4. Fine-mapped *cis*-QTLs are strongly enriched in regulatory regions across multiple cell/tissue types. (D) Enrichment of lead sQTLs within functional annotation categories from Ensembl Variant Effect Predictor (left panel), along with the proportion of all lead sQTLs falling into each annotation category (right panel). Enrichment was calculated in comparison to a background set of variants matched on MAF and distance from the TSS. Annotation categories are not mutually exclusive and therefore sum to a proportion greater than 1.

Regarding the analyses characterizing eQTL effect size differences between continental groups (he-eQTLs), we argue that there is not a clear set of analogous analyses that we could perform to characterize he-sQTLs compatible with the tools/approaches we used to discover sQTLs. Splicing is an inherently multivariate phenotype (i.e., we measure the proportion of reads that support one intron relative to other possible introns in a given splicing cluster). To allow us to use standard QTL-mapping tools, we only consider one intron at a time as a univariate phenotype when performing sQTL-mapping, rather than an entire splicing cluster as a multivariate phenotype. After sQTL mapping and fine-mapping, we merged all intron-level credible sets into gene-level credible sets. It is the lead variants from these gene-level credible sets that we use for downstream analyses. Unfortunately, it is difficult to construct an appropriate model to test for difference in sQTL effect sizes between continental groups using this set of sQTLs. Indeed, the biological interpretation of the “effect size” of sQTLs is not straightforward (i.e. there is no

widely accepted sQTL analog of aFC). We emphasize that we do explore allele frequency differentiation of our fine-mapped sQTLs (fd-sQTLs; now **Fig. S23**), and we include sQTLs in our newly added colocalization analysis (described in more detail above). We look forward to the development of tools and methods to better characterize/quantify sQTL effect sizes.

Major Comment #4

The authors depict in Figure 1 how diverse their cohort is compared to other eQTL studies such as GTEx, Geuvadis and AFGR. However, the authors failed to cite the study by Stranger et al. Plos Genetics 2012, in which they had a similar sample size and used LCLs from eight global populations of the HapMap3 project that included populations not present in GTEx, Geuvadis and AFGR. This new MAGE study uses RNA-seq and WGS instead of microarray expression and microarray genotypes, and more sophisticated analyses are made with new tools available, but the main results remain very similar to the Stranger et al study.

Thank you for alerting us to the omission of this relevant citation, which we now reference in the introduction. In addition, we have also added a citation of Storey et al., *Am J Hum Genet.* 2007 (doi: 10.1086/512017), which performed variance decomposition to show that there is greater variance between individuals than between populations, albeit with gene expression microarrays and a much smaller sample size ($n = 16$). We agree that these studies provide valuable biological context and that the Stranger et al. study achieved a similar sample size and diverse ancestry composition. The revised introduction describes the key findings from those studies and motivates our work in a more nuanced and accurate way. Specifically, the use of RNA-seq versus microarray profiling of gene expression 1) enables splicing analysis and 2) affords much higher power for QTL discovery, including characterization of widespread allelic heterogeneity. The added text is copied below in response to Referee #3 Major comment #2. We have also revised the legend of Figure 1 to clarify that it compares our dataset to other large RNA-seq-based datasets with paired whole genome sequencing data.

Minor Comments:

Minor Comment #1

In Figure 5A, there is no legend for: C, R, U

We have now placed the text: *“Allele frequencies are categorized as unobserved (U), rare variants with population allele frequencies < 5% (R), and common variants (C) with allele frequencies greater than 5%.”* into the caption for Figure 5A to more clearly define our definition for common, rare, and unobserved variation.

Referee #3: Human population genetics

In this manuscript, Taylor et al. present a new resource comprising RNA-seq data from 731 globally distributed individuals of the Thousand Genomes Project. They additionally evaluate the geographic distribution of variation in gene expression and splicing, map signals to putatively causal loci, assess the magnitude of effect sizes across populations for causal variants, and comment on the evolutionary implications of their findings. I believe this resource will be highly valuable to the field, complementing GTEx in its current use cases, and ultimately will allow for improved ancestral generalizability of future work evaluating gene expression. The researchers indeed highlight here the value of including more diverse representation in expression and splicing analyses, finding a healthy number of both eQTLs and sQTLs that are restricted to an understudied population. I believe the analyses presented here are sufficient to tell the story, but have a few questions for the authors regarding potential analytic concerns, and think that the text could use some expansion to aid the reader in appreciating the value of this work.

Thank you very much for appreciating the value of this work, as well as for the excellent suggestions for improving various analyses and their contextualization.

Major Comments:

Major Comment #1

How concerned should we be about potential reference bias? That is, mapping to existing traditional reference genomes will work better for populations that are better represented in the construction of those genome builds. How big of an impact would this be on your results, as it could lead to poorer evaluation of expression in the underrepresented populations included here?

This is a very interesting question, which has been the focus of some of our separate work and that of our colleagues. Given the relatively low levels of variation in human populations, the practical impact of reference bias tends to be small, but nevertheless important for certain sensitive applications (e.g., analysis of allele-specific expression) or in hyper-diverse regions of the genome.

Previous studies (e.g., Degner et al., 2009, *Bioinformatics*, doi: 10.1093/bioinformatics/btp579; Chen et al., 2020, *Genome Biol.*, doi: 10.1186/s13059-020-02229-3) have explored different strategies for mitigating reference bias, finding that "ancestry-matched" reference genomes yield very modest benefits. This finding is intuitive given the distribution of common genetic variation, which is largely shared across human populations. Meanwhile, rare variation will be poorly captured in an ancestry-matched reference, as by definition it is unlikely to be shared by another individual, regardless of their membership in any broad ancestry group.

To demonstrate this point in our data, we quantified the proportion of unmapped reads per sample, stratifying by population and including reads that were unmapped due to too many

mismatches or unmapped simply because no sufficiently identical reference sequence was identified (i.e., "% of unmapped reads: other"). We observe no systematic difference in the proportion of unmapped reads according to population or continental group (see figure below), despite the fact that the reference genome used in our study (GRCh38) is primarily derived from a single donor individual (RP-11) of African American ancestry.

Nevertheless, the reviewer's point is well taken, and over the longer term, we believe that pangenome/pantranscriptome-based approaches will reduce reference biases relative to the use of a single arbitrary linear reference (Sibbesen et al., 2022, *Nat. Methods*, doi: 10.1038/s41592-022-01731-9). We now mention applications of our data to pangenomics in the Discussion section:

"This expansion of diversity in functional genomics parallels efforts for improved representation of diversity in genome sequencing and assembly, including construction of pangenomes^{57,58}. While the current study was based on alignment to a linear representation of the reference genome, given the maturity of software tools and annotations built on this paradigm, MAGE offers an ideal data resource for testing pangenomic methods over the coming decade as they are developed by the research community."

In addition, we have integrated text from this response, as well as the figure below to the supplementary material (**Fig. S4**).

Figure S4. Percentage of unmapped reads per sample, stratified by population. Unmapped reads are separated into reads that were unmapped due to too many mismatches (top row) or unmapped simply because no sufficiently identical reference sequence was identified (bottom row).

Major Comment #2

The introduction is quite brief. More background on prior studies on expression across populations, eQTLs, sQTLs would be helpful to set the scene for this work and clarify the current state of the field.

We agree that additional background and context is warranted. While mindful of the constraints on length and number of references, we have expanded the introduction to describe previous RNA-seq studies in diverse cohorts in a more thorough and nuanced way, while more carefully describing the value of our study within this context. In particular, we emphasize the sample size, geographic diversity, and open access nature of the data, which enhances statistical power for understanding the distribution and sources of expression- and splice-altering variation, while also reducing barriers for functional and evolutionary genomic studies by the human genetics community.

"To this end, several studies have profiled gene expression in geographically diverse samples, offering valuable insight into the manifestation of human population structure at the level of genome function¹⁰⁻¹². These studies observed that gene expression and splicing differences between populations are relatively rare and that divergence in these molecular phenotypes does not clearly reflect patterns of population divergence. Studies also revealed an abundance of genetic variants associated with levels of gene expression (termed expression quantitative trait loci [eQTLs]) which are highly enriched near transcription start sites. Promoter proximal eQTLs possessed larger effects, on average, and tended to be shared across human populations¹¹. While foundational, these studies were generally characterized by small sample sizes and/or assayed gene expression using microarrays, limiting statistical power and resolution for molecular QTL mapping and hindering integration and comparison to modern sequencing-based data sets. Meanwhile, recent work by consortia such as MESA, GALA II, and SAGE have generated RNA-seq data from thousands of samples and include representation from African American and Latin American populations^{13,14}, but their controlled access nature poses barriers to re-use, and in some cases are restricted to disease-related research that does not include the study of genetic ancestry. "

Major Comment #3

Similarly, the discussion is again pretty succinct. In particular, expanding on the utility of the novel loci discovered here in understudied populations, how inclusion of more representative samples will improve ancestral generalizability of findings, the potential downstream impacts of this resource on improving personalized medicine, etc. would highlight the value of this resource.

We thank the reviewer for this valuable feedback; we certainly do not want to undersell the value of the MAGe resource in improving ancestral generalizability of molecular QTL findings. We

previously had a sentence in the discussion describing the benefit of diverse cohorts in identifying causal variation that may be rare in subsets of the data: *“Such consistency of genetic effects further motivates the use of diverse samples for association studies, as a common causal variant identified in one population may inform the effect of that variant in a population where the same variant is rare and association testing would be underpowered.”*

We have expanded on this observation to better convey the added value of these diverse cohorts for downstream predictive and precision medicine applications:

“Such consistency of genetic effects further motivates the use of diverse samples for association studies, as a common causal variant identified in one population may inform the effect of that variant in a population where the same variant is rare and association testing would be underpowered. Because effects tend to be consistent across populations, the functional effect should be similar in the population where the allele is rare. This has direct implications for downstream predictive applications like PRS and for personalized medicine, as inclusion of these population-stratified causal variants in predictive models should lead to more accurate and generalizable predictions⁴⁹. Thus all populations—not only underrepresented populations—benefit from the inclusion of greater diversity in genetic studies.”

In addition, we added text regarding the results of our new colocalization analyses:

“Intersection of e- and sQTLs with data from GWAS may facilitate understanding of the molecular mechanisms linking genetic variation to organismal phenotypes. Using GWAS data from the PAGE study of ancestrally diverse individuals, we identified 54 GWAS signals that colocalize with e- and/or sQTL signals. While informative and substantial in absolute number, these reflect a minority of all GWAS hits. Limited colocalization between molecular QTLs and GWAS hits is well-described, largely stemming from distinct selective pressures shaping genetic variation that can be identified (with incomplete statistical power) in the two analyses⁵⁰. GWAS hits tend to occur within genes under strong purifying selection, whereas molecular QTLs are most easily identified for genes under relaxed constraint. This is consistent with our finding that genes exhibiting strong signals of selection are depleted of MAGE eQTLs. While inclusion of additional tissues and cell types modestly increases the rate of colocalization, these qualitative observations hold even for multi-tissue studies such as GTEx⁵⁰. Despite these general limitations of colocalization analysis, our results demonstrate cases where MAGE facilitates interpretation of GWAS results, particularly in underrepresented populations. We anticipate that this utility will further improve as GWAS continue to expand to more diverse cohorts.”

Finally, as noted in response to Major Comment 1 from the same reviewer, we added some text regarding potential future uses of the data for pangenomic and pantranscriptomic studies:

“This expansion of diversity in functional genomics parallels efforts for improved representation of diversity in genome sequencing and assembly, including construction of pangenomes^{57,58}. While the current study was based on alignment to a linear representation of the reference

genome, given the maturity of software tools and annotations built on this paradigm, MAGE offers an ideal data resource for testing pangenomic methods over the coming decade as they are developed by the research community."

Major Comment #4

It was unclear to me how the presence of duplicated samples presented in the ADMIXTURE plot was handled. The methods seem to imply that one mega ADMIXTURE run of all samples was conducted, which would slightly bias ancestry estimation if some samples are included multiple times (e.g. the same YRI are showing up in MAGE, Geuvadis, and AFGR?). This shouldn't qualitatively change the results drastically, but would affect the global proportion estimates.

This is a good question. To clarify, when samples were represented in multiple studies, we only included them once in the ADMIXTURE analysis, though their results are depicted multiple times the figure (in multiple panels). We have clarified this point in the figure legend. We also note that the input genotype data from 1000 Genomes Project samples was obtained from a single source—the high-coverage (30×) sequencing study by the New York Genome Center (Byrska-Bishop et al., 2022, *Cell*, doi: 10.1016/j.cell.2022.08.004), whereas Geuvadis used an earlier version of these genotype data based on lower coverage sequencing.

Major Comment #5

The authors point out the lack of population labels in GTEx. Given their analyses here, they are able to infer ancestry for each participant, though. It would be very useful if the authors could provide population assignments for GTEx samples based on their work for use in future studies. I cannot speak to any administrative challenges in terms of interfacing with GTEx, but is this logistically possible?

Thank you for this idea for facilitating future research, which is one of the primary goals of our study. As noted in other parts of these reviews, the relationship between genetic ancestry, race, and geographically defined population labels is complex and fraught with a long history of scientific racism. As such, we are wary of assigning discrete population labels to a study where such labels were not clearly defined. We note that GTEx provides self-reported race as part of its controlled-access metadata. To address the reviewer's comment and noting the distinction between race and ancestry, we now provide principal component scores (for the top 20 PCs) for all samples displayed in Figure 1, which include 1000 Genomes samples from MAGE, Geuvadis, and AFGR as well as the MKK HapMap samples from AFGR and all samples from GTEx. We hope that these data will enhance the transparency and reproducibility of our study while also facilitating future analyses of the distribution and evolution of gene expression across diverse human populations.

Major Comment #6

The authors cite their findings as indicative of negative and stabilizing selection at different points in the manuscript. This is not wrong per se, but a bit of explanation for readers not embedded in evolutionary genomics might help unpack the interpretation of how both modes of selection can be invoked. E.g., evidence for maintenance of expression levels = stabilizing selection, but the constraint on DNA sequence underlying the eQTL used to support this argument is actually from negative selection removing sequence variation.

This is a very good suggestion. We now refer to the previously reported signatures as evidence of negative selection, removing the reference to stabilizing selection from the figure legends. However, we have also added a sentence describing the results of our analyses upon conditioning on the direction of eQTL effect. Because our results are consistent regardless of whether the minor allele increases or decreases expression of the respective genes, we interpret this as evidence of stabilizing selection, as follows:

"The shift toward smaller eQTL effect sizes in highly constrained genes holds regardless of whether the minor allele is associated with higher (Δ mean $|\log_2(\text{aFC})| = -0.284$; two-tailed Wilcoxon rank sum test: $W = 989,366$, $p = 5.58 \times 10^{-50}$) or lower expression (Δ mean $|\log_2(\text{aFC})| = -0.244$; two-tailed Wilcoxon rank sum test: $W = 1,074,750$, $p = 5.39 \times 10^{-43}$), consistent with a model of stabilizing selection whereby gene expression is maintained within an optimal range."

We hope that this more explicit test and explanation helps clarify these concepts for readers from diverse fields.

Major Comment #7

Also regarding evolution, could the authors utilize more localized selection statistics to assess the impact of negative selection on expression related loci? Currently they are just using the gene-level metric pLI, but this would miss any nuance of the particular area of a gene that might be under strongest evolutionary constraint. Zooming in to narrower areas around the causal variant to show signals of elevated negative selection would be even more convincing than a gene-wide score.

The purpose of our original analysis was to demonstrate the association between constraint against loss-of-function protein-coding sequence variation (i.e., pLI) and constraint against expression-altering variation (i.e., number of credible sets and eQTL effect sizes). We have revised the text to state this conclusion more explicitly.

Nevertheless, this is a very interesting suggestion. In seeking to address it we considered several different analyses, but one central challenge we encountered is that metrics of negative selection

based on levels of polymorphism in regions that may harbor eQTLs (i.e., near fine-mapped causal variants) will be circular, as we seek to test whether eQTLs are depleted in these very regions. To avoid this circularity, we turned to divergence/conservation data in the form of phyloP scores (based on multiple alignment of mammal genomes) averaged within intervals around transcription start sites that define putative promoter elements based on various intervals around the TSS. Consistent with the gene-level pLI results, genes with strong evolutionary conservation across species within putative promoter regions possess significantly fewer credible sets and smaller eQTL effect sizes, on average, compared to genes with less conservation in the same regions. These results are now integrated into the relevant main text of the manuscript and supplementary figure (**Fig. S14**), which are also copied below:

*"These results suggest an association between constraint against loss-of-function protein-coding sequence variation (i.e., pLI) and constraint against expression-altering variation (i.e., number of credible sets and eQTL effect sizes). This association holds for several other metrics of mutational constraint that include intolerance to copy number variation (i.e., pHaplo and pTriplo) as well as divergence-based estimates of sequence conservation in putative promoter elements (**Fig. S14**). Together, our results are consistent with previous analyses demonstrating weak, but measurable selection against expression-altering variation³³."*

A. Gene-level metrics

B. Promoter-level metrics

Figure S14. Evidence of negative selection on expression-altering variation across a range of mutational constraint metrics. **A.** Top row: number of credible causal sets for genes in (pink) and outside (blue) the top decile of various gene-level constraint metrics (pLI⁷⁹, LOEUF⁷⁹, pHaplo⁸⁰, pTripto⁸⁰, hs⁸¹, RVIS⁷⁸) obtained from the literature. Bottom row: effect sizes ($|\log_2(\text{aFC})|$) of lead eQTLs within (pink) and outside (blue) the same categories. **B.** Same as panel A, but for mean PhyloP scores summarizing conservation among genome sequence alignments of 447 mammals within putative promoter elements, defined based on intervals around the TSS ($[-1000, 1000]$ bp, $[-500, 0]$ bp, $[-50, 0]$ bp).

Minor Comments:

Minor Comment #1

Figure 1 legend – ADMIXTURE plots are described as presenting ‘hypothetical’ ancestry components. I think a better word choice could be used – inferred would be more technically correct.

Thank you for this suggestion. This change has been made.

Minor Comment #2

The dashed lines in figure 3 are not the most intuitive (the line looks like a threshold definition on first glance), though I appreciate what the authors are trying to accomplish by including them. A different way to visualize this might be better... perhaps a bracket?

Thank you for this suggestion. We have replaced the dashed line with a bracket in panel A and an arrow in panel B, which we hope the reviewer agrees is more intuitive.

Minor Comment #3

There are some font/formatting issues to be fixed.

Thank you for bringing this to our attention. We have done our best to check that all fonts and formatting are displayed as intended in the resubmitted manuscript.

Reviewer Reports on the First Revision:

Referees' comments:

Referee #1 (Remarks to the Author):

The authors have satisfactorily addressed my comments and concerns. I especially appreciate the addition of the PAGE coloc analyses and more detailed discussion. I have a couple minor suggestions that could further improve the paper's presentation:

1. In the Fig 2 violin plots, the medians/boxes do not look different for splicing. Is there a better way to plot these results? At minimum, I suggest including the statistical test results in the figure legend to show how you determined that the variance differs among populations.

2. Your GitHub repo is well organized, and I especially appreciate the READMEs describing the code used for each major analysis. The only major analysis currently missing a README when I reviewed your code is the ADMIXTURE directory, please add. I also suggest numbering each major analysis directory in the order required to reproduce your work.

Thanks,
Heather Wheeler

Referee #1 (Remarks on code availability):

I can confirm the data are publicly accessible as raw reads and results summaries.

While I did not test the code, the GitHub repo is well annotated, well organized, and includes separate READMEs for most major analyses performed, which will make a great resource for the community.

Referee #2 (Remarks to the Author):

The authors did a fantastic job at addressing my comments. I have no further comments.

Referee #3 (Remarks to the Author):

In the revised version of this manuscript, Taylor et al. have taken care to expand discussion of the nuanced topic of ancestry in multiple sections, including in revision of Figure 1 and its legend, and in the main text at several points. They also have provided more context about the added value of their new resource given the landscape of existing resources, as well as clearly laid out its limitations. I thank the authors for their thorough consideration of this and the other reviewers' suggestions and now find this manuscript suitable for publication in Nature.

Referee #3 (Remarks on code availability):

The github page is very nicely organized, and all scripts utilized across the various analyses appear to be in place.

The zenodo link in the manuscript has a typo - I manually located the MAGE code at [10.5281/zenodo.10072081](https://doi.org/10.5281/zenodo.10072081).

Author Rebuttals to First Revision:

Referee #1

Comment #1

In the Fig 2 violin plots, the medians/boxes do not look different for splicing. Is there a better way to plot these results? At minimum, I suggest including the statistical test results in the figure legend to show how you determined that the variance differs among populations.

We agree that while statistically significant, these differences are minute and thus visually imperceptible. However, we believe that this represents the most honest and transparent presentation, so we have opted to retain this version in the main text. To address the reviewer's comment, we note that **Fig. S7** depicts the estimated mean variance within each continental group (for both expression level and splicing) and the standard error of the mean, which makes the differences between the continental groups more apparent (though note the y-axis scale). Additionally, we have followed the referee's suggestion and included the statistical test results from the main text in the legend of **Fig. 2** (e.g. "Variance in expression level differs between continental groups; $p < 1 \times 10^{-10}$, one-tailed analysis of deviance.").

Comment #2

Your GitHub repo is well organized, and I especially appreciate the READMEs describing the code used for each major analysis. The only major analysis currently missing a README when I reviewed your code is the ADMIXTURE directory, please add. I also suggest numbering each major analysis directory in the order required to reproduce your work.

Thank you for making us aware of this omission. Code for the ADMIXTURE analysis has been added to the Github in the relevant directory. Additionally, the major analysis directories have been numbered, as suggested by the reviewer.

Referee #3

Comment #1

The zenodo link in the manuscript has a typo - I manually located the MAGE code at [10.5281/zenodo.10072081](https://zenodo.org/doi/10.5281/zenodo.10072081).

We provided two Zenodo DOIs in the manuscript. The MAGE published downstream data is available here: [10.5281/zenodo.10535719](https://zenodo.org/doi/10.5281/zenodo.10535719). A copy of the MAGE Github repository with analysis and figure generation code is available here: [10.5281/zenodo.10072080](https://zenodo.org/doi/10.5281/zenodo.10072080), which is a version-agnostic link to the same page the reviewer linked. As far as we can tell, both links are functioning.